# Transformer and Pre-Transformer Model-Based Sentiment Prediction with Various Embeddings: A Case Study on Amazon Reviews

**DOI:** 10.3390/e27121202

**Published:** 2025-11-27

**Authors:** Ismail Duru, Ayşe Saliha Sunar

**Affiliations:** 1R&D Department, Türk Telekom, Ankara 06103, Turkey; ismail.duru@turktelekom.com.tr; 2Department of Computer Science, University of Warwick, Coventry CV4 7AL, UK; 3Department of Computer Engineering, Bitlis Eren University, Bitlis 13000, Turkey

**Keywords:** natural language processing, customer reviews, prediction models, cross-entropy loss, categorical cross-entropy, deep learning, sentiment, transformers, pre-transformer sequence modelling, embeddings

## Abstract

Sentiment analysis is essential for understanding consumer opinions, yet selecting the optimal models and embedding methods remains challenging, especially when handling ambiguous expressions, slang, or mismatched sentiment–rating pairs. This study provides a comprehensive comparative evaluation of sentiment classification models across three paradigms: traditional machine learning, pre-transformer deep learning, and transformer-based models. Using the Amazon Magazine Subscriptions 2023 dataset, we evaluate a range of embedding techniques, including static embeddings (GloVe, FastText) and contextual transformer embeddings (BERT, DistilBERT, etc.). To capture predictive confidence and model uncertainty, we include categorical cross-entropy as a key evaluation metric alongside accuracy, precision, recall, and F1-score. In addition to detailed quantitative comparisons, we conduct a systematic qualitative analysis of misclassified samples to reveal model-specific patterns of uncertainty. Our findings show that FastText consistently outperforms GloVe in both traditional and LSTM-based models, particularly in recall, due to its subword-level semantic richness. Transformer-based models demonstrate superior contextual understanding and achieve the highest accuracy (92%) and lowest cross-entropy loss (0.25) with DistilBERT, indicating well-calibrated predictions. To validate the generalisability of our results, we replicated our experiments on the Amazon Gift Card Reviews dataset, where similar trends were observed. We also adopt a resource-aware approach by reducing the dataset size from 25 K to 20 K to reflect real-world hardware constraints. This study contributes to both sentiment analysis and sustainable AI by offering a scalable, entropy-aware evaluation framework that supports informed, context-sensitive model selection for practical applications.

## 1. Introduction

In the digital age, the rapid growth of online platforms has generated an abundance of user-generated content, particularly in the form of reviews on e-commerce sites such as Amazon. These reviews provide a valuable resource for understanding consumer sentiment, offering insights into customer satisfaction and product perception. For organisations seeking to improve customer satisfaction and refine their offerings, analysing the sentiment within this textual data is essential. Sentiment analysis, a specialised field within natural language processing (NLP), has become an essential tool for interpreting and quantifying these sentiments, enabling data-driven decision-making and customer-centric strategies.

Since its inception, sentiment analysis has progressed from basic methods to highly sophisticated techniques. Originally, sentiment analysis relied on simple algorithms that quantified the frequency of positive and negative words in texts. However, these approaches were limited in their ability to capture the nuanced semantics and contextual subtleties of language [1].

The emergence of pre-trained vector representations marked a significant leap forward, as they enabled the encapsulation of semantic information within a continuous vector space. Techniques such as Word2Vec [2] and GloVe [3] were instrumental in improving sentiment analysis models by capturing semantic relations and word context. This development addressed previous limitations, particularly in recognising contextual nuances, by focusing on the global co-occurrence statistics of words.

The field of NLP has evolved further with the introduction of transformer models, such as BERT [4] and GPT [5], which brought a new approach to implementing attention mechanisms. These mechanisms enable the model to focus on different parts of the input text, thereby capturing contextual relationships across the entire sequence. Unlike earlier models, transformer models process data in parallel and significantly improve sentiment analysis by providing higher accuracy and a deeper understanding of context [6].

These methods have long been applied to customer reviews; for example, the Amazon Review Dataset serves as a rich source of user-generated content for analysing consumer sentiment. By applying state-of-the-art artificial intelligence (AI) models to these datasets, researchers have achieved remarkable accuracy in sentiment prediction, providing actionable insights into consumer behaviour and preferences [7]. Although these experiments are scientifically valuable, the ability to accurately predict sentiment in Amazon reviews also has profound implications for business. It enables companies to better recognise and address customer concerns, tailor product offerings to consumer needs, and improve overall customer satisfaction [8].

In sentiment analysis, understanding not only model performance but also prediction confidence and uncertainty is essential, particularly when evaluating ambiguous or neutral reviews. Information-theoretic metrics such as cross-entropy loss and prediction entropy provide powerful tools to quantify this uncertainty. Entropy-based analysis has been employed in similar contexts to assess model robustness, detect overfitting, address multi-class classification, and identify cases with low confidence [9].

However, current research studies reported in the literature exhibit shortcomings in several key aspects. While static embeddings such as GloVe and FastText are well studied in traditional models, their comparative effectiveness in deep learning and transformer architectures remains underexplored. Although the overall performance of deep learning models with different architectures, such as LSTM-based, dense, CNN-based, or hybrid models, has been extensively investigated, there is a lack of research examining their strengths and limitations across different types of reviews, particularly in terms of contextual understanding and failure cases. Our study addresses this gap by providing a detailed analysis in a dedicated section.

Furthermore, existing studies often focus on traditional, deep learning, or transformer-based models in isolation, rather than systematically comparing all three within a unified evaluation framework. While several studies confirm that transformers offer high accuracy, they also require substantial computational resources. Few studies explore how to balance dataset size and computational feasibility while maintaining high performance.

Most previous research applies sentiment analysis models to general datasets, whereas this study focuses specifically on the Amazon Magazine Subscription Reviews (2023) dataset (https://amazon-reviews-2023.github.io/, accessed on 19 November 2025), thereby ensuring the relevance and robustness of our findings. The dataset used forms part of the collection gathered by McAuley Lab [10].

This paper aims to explore the evolution of sentiment analysis, from basic word frequency methods to sophisticated vector representations and the transformative impact of transformer architectures. By focusing on Amazon magazine reviews published in 2023, we evaluate the performance of state-of-the-art AI models in sentiment prediction and highlight the capabilities and limitations of current technologies in processing and interpreting large volumes of user-generated content. The research therefore seeks to answer the following research questions:Q1How do different embedding methods (Bag of Words, TF-IDF, GloVe, FastText) affect sentiment prediction performance in traditional and deep learning models as measured by accuracy, precision, recall, F1-score, and cross-entropy loss?Q2How do traditional machine learning, pre-transformer deep learning, and transformer-based models compare in terms of accuracy, cross-entropy loss, and classification confidence across varying review types?Q3What are the trade-offs between predictive performance, cross-entropy loss, and computational cost when using pre-transformer versus transformer-based models for sentiment analysis?Q4How does dataset size impact the performance and uncertainty reduction of transformer models when computational constraints are considered?Q5Which models and embedding strategies are most effective for sentiment prediction in domain-specific datasets such as the Amazon Magazine Subscriptions Reviews (2023), and how do they differ in their ability to model reviews such as ambiguous or mismatched sentiment cases?Q6Beyond overall accuracy and F1-score, how do the studied models differ in terms of entropy-based error distribution, i.e., their ability to correctly classify challenging sentiment samples while maintaining low uncertainty and misclassification loss?

## 2. Literature Review: Evolution of Word Representation and Artificial Intelligence in Sentiment Analysis

The evolution of sentiment analysis techniques in the context of Amazon review datasets highlights the transformative role of word representation in natural language processing. From initial frequency-based methods to advanced neural representations, advances in word representation techniques and the application of advanced sentiment detection tools such as TextBlob and VADER have significantly improved the accuracy and contextual understanding of sentiment analysis models, as well as our ability to analyse and interpret consumer sentiment. This review examines the methods and findings from recent research, with particular focus on how different word representation techniques have been applied to sentiment analysis and prediction in customer reviews.

### 2.1. Prioritising Based on the Frequency of the Word

In the early days of sentiment analysis, methods primarily quantified word frequency to draw conclusions about sentiment. Techniques such as Bag of Words (BoW) [11] and Term Frequency-Inverse Document Frequency (TF-IDF) [12] played a decisive role in this. Despite their simplicity, these methods were effective in highlighting key terms that contributed significantly to the sentiment expressed in texts. For example, TF-IDF was used to extract essential features for sentiment analysis in customer reviews, demonstrating its ability to identify terms critical to determining sentiment.

A slightly different method is N-grams [13], which represent contiguous sequences of n elements (words or characters) in a document. For example, unigrams (1-g) are single words, bigrams (2-g) are pairs of consecutive words, and trigrams (3-g) are triplets of consecutive words. Thus, N-gram models capture local word patterns and sequential information, providing a more nuanced representation compared to BoW. While N-grams can capture certain contextual information and relationships between neighbouring words that BoW and TF-IDF may miss, they can become impractical with larger vocabulary sizes and increased dimensionality.

These methods are used to introduce the words from a given text as input to the algorithm, which produces an output such as a predicted rating or a recommendation of a similar item. The researchers applied one or more of these methods to compare or combine them to achieve better performance.

For example, Mishra et al. [14] apply the TF-IDF method to hotel reviews to generate suitable inputs for recommendations, highlighting the model’s effectiveness in extracting meaningful features from textual data for sentiment-based filtering.

Pimpalkar and Raj [15] investigate the impact of different preprocessing techniques on the performance of machine learning algorithms by applying TF-IDF and BoW to text analysis tasks. Their insights show that preprocessing steps can significantly affect model results, with the impact of TF-IDF being negligible, while the F1 score of the classifier is significantly improved when using BoW features.

Tripathy et al. [16] investigate the effectiveness of n-gram methods in sentiment classification of movie reviews from the IMDb dataset, comparing different machine learning algorithms and emphasising the role of preprocessing and vectorisation techniques. The study shows that their algorithm performs better when using unigrams or bigrams, and that accuracy decreases as n increases, likely due to the short length of the reviews in the dataset.

To specifically compare these three methods, several studies have been conducted on different data sets. Hasan and Matin [17] compare the performance of TF-IDF with N-gram and BoW with N-gram for sentiment analysis of customer reviews. Hasan and Matin [17] also confirm the results of Tripathy et al. [16], which state that accuracy decreases significantly with higher values of n. Contrary to the findings of Pimpalkar and Raj [15], this study finds that TF-IDF performs better than BoW for most techniques, noting that this study combines them with the N-gram method.

While frequency-based methods have been effective, they process the text as a unified whole, often overlooking localised expressions of sentiment. To address this, Garapati and Chakraborty [18] propose the Review Text Granularity (RTG) model, which extracts and prioritises sentiment-rich phrases using linguistic patterns before applying traditional vectorisation. This selective emphasis improves feature quality for classifiers such as SVM and logistic regression, demonstrating how context-aware granularity can serve as a valuable preprocessing step before conventional embedding techniques.

These studies reveal the varied strengths and applications of TF-IDF, BoW, and N-gram approaches, while also highlighting the need for more semantically focused preprocessing. This paves the way for improved word representation and modelling strategies in sentiment analysis.

### 2.2. Pre-Trained Vectorisation of the Words

The introduction of pre-trained word vectorisation marked a significant advance in representing words with moderate contextualisation. Word2Vec, GloVe, and FastText were groundbreaking, enabling words to be represented as vectors in a continuous vector space. As their names suggest, these techniques represent words as continuous vectors in a multidimensional space. They usually consider the local context, focusing on words within a fixed-size window around the target word. These methods typically use shallow neural network architectures to learn distributed representations of words based on co-occurrence patterns in large text corpora. Pre-trained vectorisation enabled the capture of semantic similarities between words, but with a limitation: each word was represented by a single vector, ignoring polysemy—the existence of multiple meanings for a single word depending on context. To some extent, these models have addressed this limitation by providing a foundation for a more nuanced understanding and representation of words in text, which can be very useful in sentiment prediction tasks.

**Word2Vec**, developed by Mikolov et al. [2], uses neural networks to convert words into vector representations. The model captures semantic and syntactic relationships between words based on the context in which they appear, which can significantly improve the accuracy of classifying customer reviews as positive, neutral, or negative. Global Vectors for Word Representation (**GloVe**), introduced by Pennington et al. [3], is another influential model that generates word embeddings. Unlike Word2Vec, GloVe is based on matrix factorisation techniques applied to the word co-occurrence matrix. It combines the advantages of global matrix factorisation and local context window methods, providing rich word embeddings that can be effectively used in sentiment analysis to understand the broader context and sentiment in customer reviews.

**FastText** [19], developed by Facebook’s AI Research lab, extends Word2Vec by incorporating subword information. FastText considers not only whole words but also prefixes and suffixes, breaking words into subword units using n-grams. This allows the model to handle words outside its vocabulary, making it particularly effective for sentiment analysis of customer reviews, which often include slang (“tbh” for “to be honest”), misspellings (“amzn” instead of “Amazon”), and newly coined words (“tofurkey” for “vegan tofu turkey food”) [20,21]. By leveraging subword information, FastText ensures that the sentiment analysis model remains robust and accurate, even when faced with unconventional language.

There are studies in the literature comparing the performance of these three word vectorisation methods. For example, Dharma et al. [22] conduct a study in which they apply these three methods to generate word embeddings for a dataset of 20 news topics. The authors found that FastText performed better than the others, but the difference in performance was not statistically significant.

Kaibi et al. [23] conduct a study to evaluate the performance of different pre-trained word vectorisation methods for sentiment analysis on Twitter. The results show that FastText performed slightly better than Word2Vec and GloVe.

This is further confirmed by a study on sentiment analysis of hotel reviews. Khomsah et al. [24] compare Word2Vec and FastText. Similarly, FastText produced better results; however, the difference was negligible.

### 2.3. Pre-Transformers: Sequential Deep Learning Models for Sentiment Classification

Given the limitations of the aforementioned methods for representing words in natural language processing, researchers have increasingly focused on more sophisticated models designed to capture the dynamic nuances of language in sentiment analysis [25]. These advances have centred on the use of sequential deep learning models, particularly those based on recurrent neural networks (RNNs). By employing pre-trained vector representations of text in the embedding layers, these models have demonstrated an improved ability to capture the semantic and syntactic subtleties essential for accurate sentiment prediction. Prior to the advent of transformers, sequential deep learning models played a crucial role in interpreting the contextual and sequential aspects of language in sentiment analysis, thereby advancing the field.

Early attempts at sentiment classification often relied on RNNs and their variants, such as Long Short-Term Memory (LSTM) networks, for sequential modelling of textual data. RNNs are designed to process sequences of data and are well suited to text processing tasks where word order is important for decoding general sentiment [26]. As RNNs retain the memory of previous input, they can effectively capture long-term dependencies and contextual information in text, overcoming the limitations of fixed window size vector representations. However, practical challenges arise, especially for long sequences, due to issues such as the vanishing gradient problem, which makes it difficult to learn correlations between distant words [27]. Variants of RNNs, such as LSTM and Gated Recurrent Units (GRU), address these challenges by mitigating the vanishing gradient problem and improving the modelling of sequential data [28]. LSTM networks, equipped with memory cells, are characterised by their ability to maintain long-range dependencies and thus improve contextual meaning beyond the capabilities of vector representation models. Furthermore, the results of Ruder et al. [29] show that hierarchical bidirectional LSTMs (BiLSTMs) are more successful than LSTMs at distinguishing the sentiment of sentences such as “don’t leave the restaurant without it” or “no comparison”, where negative words are used to highlight a positive experience, in datasets covering five domains (restaurants, hotels, laptops, phones, cameras) and eight languages (English, Spanish, French, Russian, Dutch, Turkish, Arabic, Chinese) from the SemEval-2016 Aspect-based Sentiment Analysis task.

Li et al. [30], for example, conducted a comprehensive study on sentiment analysis of Amazon product reviews at the sentence level using a BiLSTM model with an attention mechanism. Achieving an accuracy rate of up to 0.96, the study demonstrated the potential of combining LSTM networks with attention mechanisms for sentiment analysis tasks. This approach uses nuanced word representations to capture emotions expressed at the sentence level, highlighting the importance of advanced word representation techniques for improving model interpretability and accuracy in sentiment analysis.

In addition to RNNs, Convolutional Neural Networks (CNNs) are frequently used in sentiment analysis. Originally developed for image processing, CNNs have been adapted for text classification, including sentiment analysis [31]. By applying convolutional operations, CNNs extract local features from input sequences, enabling the detection of hierarchical patterns in text data. These models are characterised by their ability to identify local patterns, such as key phrases or word combinations that indicate a particular mood [32]. For example, Tammina et al. [33] design a CNN-based deep learning model to predict the sentiment of embedded sentences using the Word2Vec method. The authors used IMDB movie and Amazon product reviews as datasets. The proposed model significantly outperformed traditional machine learning methods; however, the accuracy of 68% on the Amazon dataset and 74% on the IMDB dataset is not particularly impressive.

Similarly, Kumari et al. [34] used Amazon Product Reviews to compare various deep learning models, including RNN- and CNN-based architectures. Their findings also confirmed that while many of these models perform well, none demonstrated exceptional performance.

The researchers also combine the above-mentioned models to create hybrid models for more accurate results. For example, Habbat et al. [35] investigate various deep learning designs combined with different word embedding methods, including MultiFit, CamemBERT, and XLNet. The authors develop a model that combines GRU and CNN deep learning models and uses XLNet for word embedding to predict the sentiment of text input. Their model achieved the highest performance (96.5%) on the French Amazon Customer Reviews dataset.

Comparative analyses of sequential models have highlighted differing performances across datasets and domains, emphasising the crucial role of model architecture, dataset characteristics, and feature representation in determining the success of sentiment classification approaches [26,27].

#### 2.3.1. Encoders and Decoders

One of the most commonly used architectures in sequence-to-sequence models is the encoder-decoder architecture. Sequence-to-sequence models are particularly well suited to advanced natural language processing tasks such as machine translation [36], text summarisation [37], and speech recognition [38]. This architecture processes and encodes the input sequence into a fixed-length context vector, capturing the essential information before decoding it step by step to generate the output sequence. The context vector serves as a summary or representation of the input sequence. At each step, the decoder attends to different parts of the input sequence, guided by an attention mechanism, to generate the next token in the output sequence. The encoder-decoder architecture has proven highly effective in capturing the semantics and structure of sequential data, enabling accurate and contextual predictions [39].

#### 2.3.2. Attention Mechanism

To improve model performance and reduce computational burden, the attention mechanism has been adopted in encoder-decoder architectures. This modified mechanism is inspired by the human visual system, which continually focuses on small parts of a scene until it is fully recognised [40]. Since 2015, the use of attention mechanisms has become increasingly popular in natural language processing and object recognition [40]. In the context of attention mechanisms in deep learning, the main idea is to assign weights dynamically to different parts of a fixed vector representation, such as the hidden states of the encoder in sequence-to-sequence models, depending on the context of the current step in the decoding process [41]. By assigning these weights, the model learns to focus on specific parts of the input sequence, rather than using the entire vector representation uniformly. The focus is determined by the type of attention. Some studies (e.g., [42]) use sentiment as a weighting factor in the attention mechanism to detect fake news. There are several types of attention based on softness, such as soft or global; forms of input, such as item-wise or location-wise; input representation, such as self-attention or hierarchical; and output representation, such as single-output, multi-head, or multi-dimensional [43].

Despite their successes, sequential deep learning models face challenges such as handling very long sequences and the computational complexity of training. Furthermore, the emergence of transformer models has shifted the focus, as these models are better able to manage long-range dependencies through self-attention mechanisms and parallelise computation.

### 2.4. Transformer Models for Enhanced Word Representation

In 2017, Vaswani et al. [44], in their paper titled ’Attention is All You Need’, present an idea centred on the attention mechanism, that you have probably encountered many times before. Recall briefly from the previous section that RNNs focus on sequential data at each timestamp or each subsequent sentence in a sequence. However, what Vaswani et al. [44] propose is designed to process sequential data such as text for tasks including translation, summarisation, and text generation, without the need for recursion or convolution.

The core idea behind transformers is the attention mechanism, specifically **the mechanism of self-attention**. This enables the model to weight the importance of different words within a sentence, regardless of their positional distance from each other [45]. This allows the model to focus on all parts of the input sequence simultaneously, making it highly efficient for parallel computation and able to capture complex linguistic structures and dependencies, which reduces ambiguity. The self-attention mechanism tracks the relationships between words within the input and output sentences to calculate how similar each word is to all other words in the sentence, including itself [43].

To use positional information to highlight the relationships between words in a sentence, **positional encoding** is proposed by transformer models. This mechanism assigns weights to words based on their positions in the sequence. This enables parallel processing, overcoming the computational overhead caused by training RNNs. For example, Cao et al. [46] propose an improved BERT model by adding a position vector and a segment vector so that the input texts and the positions of the input words can be determined without storing them in the memory of RNNs to analyse the sentiment of agricultural product reviews.

Wang et al. [47] modified the idea of the introspection mechanism for sequences. The authors argue that the sequence of sentences in a customer review is not always meaningful. The first and last sentences might be coherent, while the sentences in the middle are not always sequentially related. In their study, instead of sentence-to-sentence attention, they consider the relationships between sentences in the document vector space and apply self-attention according to the global average pooling of sentence representations. The proposed model was applied to a set of Amazon customer reviews data, and the results are consistent with the baseline accuracy established by other researchers.

Transformer architectures differ depending on how they are structured around the encoder and/or decoder. The first encoder-only model, and one of the best-known transformer models, is BERT. Encoder-only models can solve many natural language understanding tasks, such as classification, sentiment analysis, summarisation, and others. BERT has been modified several times to improve performance and efficiency, resulting in variants with different names, such as DistilBERT, RoBERTa, XLM, ALBERT, ELECTRA, and many others, due to changes in architecture.

Kokab et al. [48] use hybrid BERT and deep learning architectures. In their study, BERT’s zero-shot classification is used to label reviews by calculating their polarity scores. A pre-trained BERT model is then used to obtain sentence-level semantic and contextual features from this data and generate embeddings. A neural network is subsequently used to extract semantics from the contextual embedded vectors. For comparison, the models are applied to the United States presidential election and IMDB datasets, along with models using Glove and Word2vec for embeddings, and CNN, LSTM, and Co-LSTM layers as the main model. While all results are generally very promising, the model using BERT with embeddings generally outperforms the others. However, in some cases, the LSTM model with embeddings from Word2vec and the Co-LSTM model with embeddings from Glove showed better results in terms of precision, even though their accuracy is lower.

Decoder-only models, such as OpenAI, GPT and its versions, are usually used for text generation tasks where predicting the next line or word is important.

The first encoder-decoder model is T5 (Text-to-Text Transfer Transformer), which takes an input text and, through decoding, generates a label as a prediction for text classification. Another example is BART, which combines BERT and GPT.

In summary, unlike the limitations of fixed-size vector representations and sequential input training in neural networks, advanced word representation models capture contextual information from entire text sequences using transformer architectures. Rather than representing individual words, these models represent whole sentences or sequences of tokens as contextual embeddings. The integration of sentiment-aware pre-training models and fine-tuning strategies with advanced embedding techniques such as DeBERTa, mBERT, ALBERT, and ELECTRA has shown promising results in achieving higher accuracy and a more sophisticated understanding of text sentiment.

As transformer models and large language models are relatively recent innovations, an increasing number of studies are examining their effectiveness in sentiment analysis using Amazon product review datasets. For example, Ali et al. [49] conduct a comparative analysis similar to ours, but with a more limited set of models and embedding techniques. Similarly, Durairaj and Chinnalagu [50] investigate a fine-tuned BERT model for sentiment classification across multiple datasets, including Amazon. However, the application of transformer-based approaches in this context remains insufficiently explored, indicating a clear gap in the literature. Our study addresses this gap by systematically evaluating a diverse range of models, from traditional machine learning to pre-transformer deep learning and transformer-based architectures, alongside various embedding strategies, spanning frequency-based static methods to dynamic contextual embeddings.

### 2.5. Information-Theoretic Metrics in Sentiment Analysis

In sentiment classification, traditional evaluation typically relies on accuracy-based metrics such as accuracy, precision, recall, and F1-score, which quantify the proportion of correctly or incorrectly classified instances. While these metrics are useful for measuring overall performance, they fail to capture the reliability or confidence of model predictions, particularly in the presence of ambiguous, noisy, or user-generated text. To address these challenges, researchers have increasingly adopted information-theoretic metrics, which provide deeper insights into prediction uncertainty and model calibration.

Cross-entropy loss is one of the most widely used metrics in deep learning for classification tasks, including sentiment analysis. Sepúlveda-Fontaine and Amigó [9] examined the role of entropy in machine learning. The authors identified cross-entropy as a key metric used in deep learning, feature selection, image classification, and multi-class classification tasks. Cross-entropy measures the divergence between the predicted probability distribution and the true label distribution. Lower cross-entropy values indicate that the model not only predicts the correct class but does so with higher confidence, making it a valuable complement to conventional metrics. For example, Mu et al. [51] integrate cross-entropy and Kullback-Leibler (KL) divergence losses in multimodal sentiment analysis to improve calibration and robustness in noisy social media environments, particularly customer reviews (e.g., [52,53]). In our study, cross-entropy loss is leveraged to assess model calibration and uncertainty handling across different model paradigms: traditional machine learning, pre-transformer deep learning, and transformer-based models.

Closely related is the concept of prediction entropy, which quantifies the distribution of probability mass across possible output classes. A high-entropy output implies model uncertainty, where no single class dominates the prediction, while low entropy indicates a confident, focused prediction. In the context of deep learning and transformer models, entropy serves as a proxy for representational efficiency, capturing the model’s ability to compress linguistic information and produce decisive outputs, even under ambiguous conditions. Herrera-Poyatos et al. [54] emphasise the importance of uncertainty-aware evaluation frameworks in LLM-based sentiment analysis, highlighting that stochastic variability, prompt sensitivity, and noisy input can compromise reliability without entropy-aware calibration.

Despite their utility, few studies systematically integrate these metrics across different modelling paradigms. Existing work often focuses on accuracy-driven comparisons, overlooking the role of uncertainty in real-world applications where misclassification can have significant implications. By incorporating cross-entropy and related measures into our experimental design, this study addresses these shortcomings and contributes to the development of scalable, uncertainty-sensitive sentiment analysis approaches. Including cross-entropy in our evaluation allows us to systematically examine each model’s ability to minimise predictive uncertainty during training and testing. Furthermore, by combining entropy-based measures with qualitative error analysis (see Section 6), we bridge quantitative evaluation with interpretability, highlighting how entropy reduction correlates with the models’ capacity to resolve ambiguity in real-world sentiment data.

**The objective of this paper** is to comprehensively evaluate the performance of various learning paradigms, ranging from traditional machine learning models to pre-transformer deep learning architectures and transformer-based models, on the sentiment analysis task using Amazon customer reviews of magazine subscriptions. A central focus is to assess the impact of different embedding techniques, including frequency-based static methods such as Bag of Words and TF-IDF, pre-trained word embeddings such as GloVe and FastText, and contextual embeddings used by transformer models. In addition to performance evaluation, this study aims to identify the types of reviews where the models encounter the most difficulties, thereby providing deeper insight into the models’ limitations. By systematically comparing these approaches under consistent experimental conditions, our work highlights the strengths and weaknesses of each modelling and embedding combination. The results should assist researchers and practitioners in selecting suitable strategies for real-world applications of sentiment analysis, particularly in e-commerce and customer feedback systems. Ultimately, this study contributes to the growing literature by providing detailed, multi-level benchmarking and revealing trade-offs in performance that are often overlooked in isolated evaluations. The next section, Section 3, describes the adopted methodology in greater detail.

## 3. Methodology

We improved the methodological approach presented by Alqahtani [55]. The methodology followed is shown in Figure 1.

### 3.1. Dataset and Tools

The Amazon Reviews dataset, published in 2023, was used in this study. This dataset comprises several distinct subsets across various categories, such as All Beauty, Appliances, Electronics, Magazine Subscriptions, and Video Games. These subsets vary in the number of reviews and overall size. We selected the magazine subscription sub-dataset for this study as it is one of the smallest yet rich datasets.

The Amazon Magazine Subscription 2023 dataset, as the name suggests, stores data regarding magazine subscriptions sold on Amazon. The dataset includes two sub-datasets: a review dataset containing user reviews, and a metadata dataset providing information about the items. The main dataset used in this research is the Amazon magazine subscriptions reviews data, which has 10 features: rating, title, text, images, asin, parent_asin, user_id, helpful_vote, verified_purchase, and timestamp. It contains over 71,500 reviews with ratings for 3400 products written by 60,100 users.

We used Google Colab (https://colab.research.google.com/ accessed on 19 November 2025) as the main development environment. To complete the implementation, we used a selection of Python 3.13.5 libraries. The libraries used in this study include, but are not limited to: pandas, json line, scikitlearn, Keras, TensorFlow, transformers, and matplotlib.

We initially used 25,000 instances from the data to obtain a balanced distribution across different sentiment classes, which will be explained in later sections. However, we reduced the dataset to 20,000 randomly selected records to ensure it remained within the computational limits of Google Colab’s free graphics processing unit (GPU) quota, allowing us to complete the sentiment analysis predictions using transformer models. To further reduce the dataset size, we randomly removed reviews from both sentiment categories, ensuring class balance was preserved throughout, and thus the dataset remained free from sentiment bias. The process is visualised in Figure 2.

Although this reduction was driven by hardware constraints rather than experimental design, it reflects a realistic scenario faced by many researchers and practitioners, particularly in academic or resource-limited settings without access to high-performance computing. Therefore, this decision also highlights the importance of developing sustainable and accessible AI practices in constrained computational environments.

### 3.2. Sentiment Classification and Balancing Data

Since it is not cost-efficient for a human to read each comment to label its sentiment, we follow one of the widely applied methods as in  [56]. If a review is rated 3 stars, we exclude it from the data as it is considered neutral. If a review is rated 4 or 5 stars, it is categorised as positive; otherwise, it is categorised as negative. We do not use a neutral class (a separate class for 3-star reviews) as this reduces accuracy.

Although this approach is widely accepted in the literature, we conducted a quick experiment during the training and testing of our models. When we introduced a three-class classification setup by including 3-star reviews as a neutral category, the overall accuracy and performance of the models declined significantly. For instance, while the Random Forest model trained with BoW embeddings achieved an accuracy of 0.89 in the binary classification setting, its accuracy dropped by more than 10% when the neutral class was added.

Since most reviews are rated 3 stars or higher, almost 80% of the comments are positive, resulting in imbalanced data. To address this, we use the under-sampling method by randomly selecting instances from the positive and negative categories in our data.

In the next section, we present the results of our exploratory data analysis and describe the preparation of the cleaned and balanced data for implementing our predictive models.

## 4. Data Exploration: Descriptive Analysis of the Data

There are originally 71,497 non-null raw instances in the Amazon Magazine Subscriptions dataset. The reviews are accompanied by a rating ranging from 1 to 5, indicating how customers evaluate the product. Figure 3 shows the distribution of the data across the rating values.

Another feature in the data is helpful_vote, which indicates how many people found a particular review helpful by clicking the helpful button. The value for helpful_vote ranges from 0 to an integer maximum, which is 2169 in our dataset. We explored the general trend in the number of helpful votes a review attracts. Figure 4 shows the distribution of reviews that have received helpful votes, ranging from 0 to 100. It shows that the majority of reviews, over 40,000, have not received any helpful votes. The number of reviews that received more than 20 helpful votes is also negligible.

Figure 5 visualises the average number of helpfulness votes received by reviews, grouped by their star rating. To ensure meaningful analysis, reviews with zero helpful votes were excluded, focusing the results on those that elicited some user interaction. A clear negative correlation is observed between rating and average helpful votes, indicating that lower-rated reviews tend to receive significantly more user engagement in terms of helpfulness votes than higher-rated reviews. This suggests that critical reviews are perceived as more useful by users.

The same trend is also demonstrated by another visualisation in Figure 6. The bar chart displays the average rating of reviews grouped into bins according to the number of helpful votes, ranging from ‘Very Low’ (1–3 votes) to ‘Extremely High’ (603–2000+ votes). The chart shows an inverse relationship between the number of helpful votes and the average rating. Reviews with ‘Very Low’ helpful votes have the highest average rating of 3.51, while those with ‘Extremely High’ helpful votes have the lowest average rating of 1.76. This suggests that highly helpful reviews tend to be more critical, as users seem to value detailed feedback that highlights issues or shortcomings.

To explore whether there is a relationship between ratings and helpful votes, we visualise their changes over the years. Figure 7 and Figure 8 provide complementary insights into Amazon review engagement over time.

Figure 7 presents a bubble chart showing ratings over time, with bubble size and colour representing the number of helpful votes. The chart reveals a trend in which 1-star reviews generally receive higher average helpful votes, indicating their perceived value in providing critical insights. Clusters of large bubbles, particularly between 2015 and 2020, indicate periods of concentrated high engagement with negative reviews.

Figure 8 complements this by illustrating cumulative helpful vote distributions across ratings over time. While 5-star reviews dominate in volume, 1-star reviews contribute disproportionately to user engagement, particularly during peak periods such as 2018. Together, these figures show that high engagement does not correlate directly with high ratings; instead, lower-rated reviews generate more interaction, likely due to their perceived informational value. This underscores the importance of nuanced sentiment understanding in review analysis, especially for model training and user-centric design. In both figures, it is clear that 4-star reviews receive relatively little attention, suggesting they may be seen as less informative or less emotionally charged by other customers. Although 2-star reviews follow a similar trend, they attract slightly more engagement than 4-star reviews.

These visualisations indicate that while 5-star reviews offer volume and social proof, 1-star reviews provide actionable insights, making them highly engaging. The peak around 2018 likely reflects changes in platform behaviour, which is evident in both visualisations. As the aim is to predict review sentiment based on ratings, the helpful vote metric appears to have potential value in improving predictive models.

As an initial step in sentiment analysis, we created word clouds for reviews with different ratings after cleaning the data and removing the most common uninformative words, such as magazine, Amazon, and subscription. Figure 9a shows that the most frequently used words in 1-star reviews are time-related terms such as week, month, year, and issue. Negative sentiment is often expressed through words such as disappointment, terrible, worst, and bad. Additionally, service- and quality-related terms feature prominently, including never received, advertisements, waste of money, waste of time, longer, kindle, quality, customer service, credit card, and old. We also observed frequent use of words that may accompany negative statements, such as even, never, wouldn’t, won’t, can’t, problem, instead, and cancel.

Figure 9b shows that words associated with positive emotions, such as love, great, excellent, happy, wonderful, awesome, and best are the most frequently used in 5-star rated reviews. Positive sentiment is also conveyed through words and phrases suggesting a positive experience, such as look forward, favourite, fun, definitely, informative, parent tip, great tip, appreciate, well written, great price, interesting, and continue. Some words appear in both 1-star and 5-star rated reviews but with different frequencies. For instance, terms related to advertisement occur more often in 1-star reviews. Both categories include words related to the magazine’s genre, such as photography, recipe, history, politics, fashion, and guide, but these appear less frequently and with less variety in the 1-star reviews.

After completing the exploratory data analysis, we concluded that some features, such as helpful votes and the text data, could be valuable for creating a model to predict the sentiment of a review. In the next section, we describe the feature extraction steps we performed.

### Extracted Features

In this study, several features were extracted from the Amazon review dataset to enhance the predictive power of the sentiment analysis models. The *text*, *helpful vote*, and *verified purchases* fields were already available in the dataset. The other features listed below, which include both basic attributes derived from the review text and sentiment scores from external tools, were extracted after preprocessing. The available and extracted features are as follows:

**Text:** The main body of each review, containing customer feedback in unprocessed form.

**Helpful Vote:** The number of users who marked the review as helpful. This feature provides insight into the perceived usefulness of the review content.

**Verified Purchase:** A binary indicator showing whether the reviewer purchased the product through Amazon, which can lend credibility to the sentiment expressed in the review.

**Sentiment from Rating:** A basic sentiment classification derived from the star rating given in the review. Reviews rated 4 and 5 stars are classified as positive (1), while reviews rated 2 and 1 stars are classified as negative (0). This provides a baseline sentiment label for the model.

**Word Count after Preprocessing:** The number of words in the review text after preprocessing steps such as removing stopwords, punctuation, and irrelevant content. This feature can help capture the review’s verbosity and provide context for sentiment strength.

**Character Count after Preprocessing:** The total number of characters in the processed review text, which may correlate with the review’s level of detail and sentiment.

**VADER Compound Score:** The compound score generated by the VADER (Valence Aware Dictionary and sEntiment Reasoner) sentiment analysis library, representing the overall sentiment polarity of the text. This value ranges from −1 (most negative) to +1 (most positive).

**VADER Positive Score:** The positive sentiment score calculated by VADER, indicating the degree of positivity in the review, ranging from 0 to 1.

**VADER Negative Score:** The negative sentiment score calculated by VADER, indicating the degree of negativity in the review text, ranging from 0 to 1.

**VADER Neutral Score:** The neutral sentiment score from VADER, capturing the proportion of the text that is neutral, ranging from 0 to 1.

These features were selected to provide a comprehensive view of each review’s sentiment and structural characteristics, helping the machine learning models understand both the content and tone of the feedback. Together, they form the basis for a more accurate and nuanced sentiment prediction process.

Figure 10 shows the correlations between various extracted features from the dataset, excluding ratings, to provide a deeper understanding of their relationships. We also excluded the number of characters, as it is strongly correlated with the number of words, and selected only one for clearer visualisation. There is a moderate positive correlation between the VADER compound score and sentiment from rating (0.49), confirming alignment between sentiment scores derived from ratings and textual sentiment polarity. Another notable positive correlation is between the VADER compound score and the VADER positive score (0.61), shown in the upper left of the graph, indicating a relationship between overall positivity and specific positive sentiment values.

Interestingly, helpful votes show weak correlations with most features, including sentiment scores and text length, suggesting that helpfulness is influenced by other factors, possibly related to the content’s informativeness or critical nature, rather than straightforward textual attributes. The VADER negative score has a moderate negative correlation with both the VADER compound score (−0.60) and sentiment from rating (−0.37), reinforcing the idea that more negative sentiments detract from overall positivity. This aligns with earlier observations that critical reviews (lower ratings) received more helpful votes.

When these findings are integrated with previous analyses, it becomes clear that helpful votes are not simply associated with numerical features such as text length or compound sentiment scores. Instead, as shown earlier, reviews with lower ratings (e.g., 1-star) and more critical insights consistently receive more helpful votes, reflecting the value users place on actionable and detailed feedback. The weak correlation between helpful votes and sentiment features in the heatmap further supports this, indicating that user engagement with reviews depends more on the contextual informativeness of the review than on general sentiment polarity or text structure. This underscores the nuanced nature of user interactions with reviews and the importance of incorporating features that capture criticality and relevance into predictive models for sentiment analysis.

## 5. Results: Comparison of the Sentiment Prediction Performance of Models Categorised as Traditional Machine Learning, Pre-Transformers, and Transformers

This section is divided into three subsections to present the performance of the models using different approaches: traditional machine learning, pre-transformer, and transformer models, on sentiment prediction with the Amazon Magazine 2023 review (https://amazon-reviews-2023.github.io/, accessed on 19 November 2025) datasets.

### 5.1. Performance of Traditional Machine Learning Models with Various Embedding Methods

In this section, we present the results of applying traditional machine learning models to the sentiment prediction task on the dataset. These methods, including Logistic Regression (LR), Support Vector Machines (SVM), Decision Tree (DT), Random Forest (RF), and Gaussian Naive Bayes (GNB), served as benchmarks against more advanced models such as pre-transformers and transformers.

Before presenting the sentiment prediction results, we provide further details on the embedding vector size and external features, as summarised in Table 1.

An extended timing analysis was conducted using different experimental settings. For example, training with BoW and TF-IDF took almost an hour, whilst training with FastText was the slowest experiment, taking 21 min to train. The breakdown of the FastText experiment is as follows:Traditional ML with Only External Features (FastText Experiment):–Less than 1 min for all models (without hyperparameter tuning)–Nearly 5 min for Random Forest (with hyperparameter tuning)Traditional ML with Only FastText Vector Features:–Less than 1 min for all models (without hyperparameter tuning)–Nearly 20 min for Random Forest (with hyperparameter tuning)Traditional ML with FastText Vector Features + External Features:–Less than 1 min for all models (without hyperparameter tuning)–Nearly 21 min for Random Forest (with hyperparameter tuning)

The choice of evaluation metric depends heavily on the dataset, such as whether the splits of instances by category are balanced or normally distributed. Standard metrics for evaluating supervised models, including accuracy, enable comparative analysis of their effectiveness in handling large-scale text data with varying sentiment polarities. Accuracy reflects the overall percentage of correct predictions; precision indicates the proportion of predicted positives that are truly positive; recall measures the model’s ability to identify all actual positives; and the F1-score provides a balanced metric combining precision and recall. These metrics offer an interpretable framework for comparing the effectiveness of different models, including traditional machine learning, pre-transformers, and transformers, in predicting sentiment from the Amazon Magazine review dataset.

In this analysis, we primarily use accuracy as the evaluation metric because the dataset was balanced using under-sampling methods, resulting in an equal split across categories. This ensures that the difference between accuracy and F1-score is negligible, making accuracy a valid and straightforward choice. We present both metrics. However, if the dataset were imbalanced, the F1-score would be a more appropriate metric, as it accounts for both precision and recall, offering a more reliable measure of model performance in such cases.

Table 2 and Table 3 present the accuracy and F1-score results of traditional machine learning models for sentiment prediction, using various embedding methods (FastText, GloVe, TF-IDF, Bag of Words), as well as only normalised text with basic NLP techniques without specific embedding methods, and configurations (“Reviews Only” and “Reviews Only w/Features”).

As the overall VADER score and the positive VADER score were correlated with the ratings, we considered whether to include the VADER component scores, as their influence could potentially bias the model’s predictions. The results presented in Table 2 and Table 3 exclude the VADER component scores. However, we also conducted experiments including the VADER component scores. While some results showed a slight improvement of 1–2% in accuracy, the accuracy never exceeded 90%, indicating that the general trends in the results remained consistent regardless of whether the VADER component scores were included.

The findings offer valuable insights into the strengths and limitations of traditional approaches to sentiment analysis. In addition, we used several tuning techniques, including hyperparameter tuning, to further enhance the effectiveness of the models, particularly the Random Forest model.

This tuning led to a 5% improvement, achieving an accuracy of 0.70 when using only numeric features, without incorporating embedded review text. Other optimisation techniques, such as normalisation, were also applied to improve model performance. The highest accuracy for each model reflects the implementation of various optimisation strategies, some of which are not explicitly detailed in this paper.

As shown in Table 2 and Table 3, FastText outperforms TF-IDF and Bag of Words, achieving higher accuracy across almost all models, which confirms much of the research discussed in Section 2.1. Surprisingly, GloVe performs the worst among all models. This result highlights the effectiveness of FastText in capturing semantic relations, especially in informal and noisy text environments such as Amazon reviews. Unlike GloVe, which represents each word with a single fixed vector derived from global co-occurrence statistics, FastText incorporates partial word information by learning representations from character-level n-grams. In this way, words not in the vocabulary, misspellings, and morphological variations common in user-generated content can be better accounted for.

Interestingly, even traditional frequency-based methods such as BoW and TF-IDF outperform GloVe in several configurations. One possible explanation is that BoW and TF-IDF treat documents independently and directly emphasise sentiment-carrying terms through their frequency or term weighting. Although these simpler methods are less sophisticated, they can still be very effective in well-preprocessed, domain-specific datasets. In contrast, because GloVe relies on learned co-occurrence patterns, sentiment signals can be diluted when it is applied in simpler models that lack the architectural capacity to fully utilise these global embeddings.

Although GloVe underperforms in this particular implementation, its continued relevance in pre-transformer and transformer models is confirmed by the literature to date. The dense semantic vectors it produces can be better exploited by advanced architectures capable of modelling long-range dependencies and contextual nuances. Thus, while our results highlight the benefits of FastText and the robustness of traditional lexical methods in lightweight models, they also show that the advantages of GloVe embeddings become more apparent in deep learning contexts where model complexity matches the richness of the embedding space.

Furthermore, incorporating supplementary features, such as helpful votes, significantly enhances model performance across all embedding methods. For instance, Decision Tree shows a notable accuracy increase, rising from 0.63 without embeddings to 0.77 with FastText embeddings and features. Among the models, Logistic Regression, Random Forest, and SVM provide the most robust performance, reaching peak accuracies of 0.88 with FastText embeddings and features. In contrast, Naive Bayes models perform poorly, failing to leverage advanced embeddings effectively.

These findings highlight the importance of embedding selection and feature engineering in improving traditional machine learning models for sentiment analysis. The results also indicate that combining FastText embeddings with supplementary features is the optimal configuration for achieving higher accuracy with traditional approaches.

### 5.2. Performance of Pre-Transformer Sequential Deep Learning Models with Various Embedding Methods

The deep learning models differ in their architecture, incorporating various layers and parameters to optimise performance. We developed two sequential and one hybrid deep learning model, utilising GloVe and FastText for vector representation of the review text data.

We specifically chose GloVe and FastText as embedding methods because, as discussed in the literature review, they are more advanced and effective than traditional approaches such as TF-IDF and Bag of Words (BoW) in deep learning applications. These methods provide richer semantic representations, making them better suited to capturing contextual nuances in text. Consequently, this section focuses on applying these advanced embedding techniques in conjunction with the developed pre-transformer deep learning models, representing a progression beyond traditional machine learning algorithms.

The structures of the developed models are shown in Figure 11. The simplest deep learning model we developed is Model 1, illustrated in Figure 11a, which consists of a single Bidirectional LSTM layer following the embedding layer. We use a Global Max Pooling layer to extract the most significant features across the sequence, representing the key characteristics of the sentences in a review, such as overall sentiment. A dense layer transforms the pooled features into binary classification outputs (positive and negative). Additionally, a dropout layer randomly deactivates a proportion of neurons during each training iteration, reducing the risk of overfitting and improving the model’s generalisation. Although this simple model is computationally efficient, its structure may miss some nuanced features present in the text.

To address this limitation, we constructed Model 2, shown in Figure 11b, which stacks two Bidirectional LSTM layers. This deeper structure is better suited to capturing long-term dependencies in the data, allowing the model to retain and process contextual nuances. However, stacking multiple LSTM layers may increase the risk of overfitting, particularly with limited training data, necessitating careful use of regularisation techniques. We have also added a dropout layer to reduce the risk of overfitting in this module.

Finally, we developed Model 3, shown in Figure 11c, a hybrid structure that combines Convolutional Neural Networks (CNNs) with LSTM layers. In this model, a Conv1D layer followed by a Max Pooling layer is applied before the two stacked Bidirectional LSTM layers. The CNN extracts local patterns in the text, such as key phrases or n-grams, while the LSTM layers focus on global, sequential dependencies. Although this structure is more computationally demanding, it aims to capture both local and global text features more effectively.

Before presenting the sentiment prediction results, Table 4 provides an overview of model complexity and training time for various pre-transformer deep learning architectures using GloVe and FastText embeddings. The number of parameters, memory size, and training durations for five epochs are included to highlight computational efficiency and resource implications. All experiments were conducted on Google Colab with a T4 GPU instance.

To evaluate the performance of the developed pre-transformer deep learning models, the cross-entropy loss function is used alongside accuracy, precision, recall, and F1-score. In prediction tasks, the loss function guides model training by measuring the difference between the model’s predictions and the actual labels. This metric not only drives training but also provides insight into model uncertainty and confidence in predictions. For instance, if a model predicts a negative sentiment when the true sentiment is positive, the loss function returns a high value, indicating a significant error. The model updates its weights to minimise this loss during training, aiming to make more accurate predictions. Lower cross-entropy indicates better calibration of the model’s probability outputs and lower uncertainty. Loss functions were not use in traditional machine learning models because their training process is more straightforward and does not require the same iterative weight adjustment as deep learning models.

All three models, Simple LSTM-based (Model #1), Improved LSTM-based (Model #2), and Hybrid CNN-LSTM-based (Model #3), achieve consistent accuracy of approximately 0.89, regardless of the embedding method. This suggests that both GloVe and FastText are similarly effective at capturing the semantic information required for sentiment prediction in this dataset. However, the loss values vary slightly across the models and embedding methods, indicating differences in the optimisation process and the ability to minimise prediction error. Overall, these loss differences highlight the nuances of optimisation, with FastText achieving the lowest loss metric (0.28) in the hybrid architecture, while GloVe achieves approximately 0.31 with the two-layer LSTM configuration. When analysing the precision and recall values, we observe that several models using FastText embeddings consistently achieve strong recall values of 0.91, especially the Simple LSTM and Improved Stacked LSTM models. This indicates that these models capture most relevant sentiment cases effectively and minimise the number of false negatives. However, the Hybrid LSTM-CNN model stands out for the opposite trend. It achieves the highest precision (0.90) among the FastText-based models, but at the expense of lower recall (0.85). This pattern suggests that, while the Hybrid model is more selective and makes fewer false-positive predictions, it sacrifices its ability to identify all relevant sentiment cases. In practice, this model may be preferable when false-positive predictions are more costly, whereas the other FastText models may be more suitable when capturing all relevant cases is a priority.

### 5.3. Performance of Transformer Models

In the previous section, we developed LSTM-based and hybrid deep learning models using various types of embeddings to provide a vectorised representation of the text before inputting them into the neural network. However, the way transformers embed text and use it to extract meaning and nuances differs from using pre-trained and fixed embeddings, as explained in Section 2.4. Implementing transformer models is computationally demanding, and the increased computational cost led us to revisit the data sampling process, as the initial dataset was too large to process efficiently. The under-sampling method used to manage the dataset size is already detailed in Section 3.1.

Transformer models such as BERT and DistilBERT do not require explicitly defined embeddings like GloVe or FastText. Instead, transformers use contextual embeddings that are learned dynamically during training. The key difference is that embeddings like GloVe and FastText are static, meaning a word has the same representation regardless of its context, whereas transformers produce context-dependent embeddings during training. As Yenicelik et al. [57] point out, the word “bank” will have different embeddings in the contexts of “river bank” and “bank account,” accurately capturing the semantics. These embeddings are generated by the transformer’s self-attention mechanism, which allows the model to adjust the representation based on surrounding words, making them much more effective for tasks such as sentiment analysis, where context is critical.

In this section, we have fine-tuned four models: DistilBERT-base-uncased, BERT-base-cased, BERT-base-uncased, and RoBERTa. Although all are BERT-based models, they differ in size, primary target, and embedding method used.

The difference between the cased and uncased versions is whether the model considers capitalisation when embedding. In our context, capitalisation is usually not very important. Uncased models convert all text to lower case, ensuring that upper and lower case (e.g., “shopping” vs. “Shopping”) have no influence on the embedding, while cased models distinguish between upper and lower case forms of the same word.

DistilBERT is a smaller and more memory-efficient version of BERT that retains 97% of BERT’s performance but is 40% smaller [58]. This makes it computationally efficient when use the free GPU provided by Google Colab. RoBERTa removes the next sentence prediction feature and is trained on a larger dataset to focus specifically on understanding sentence-level semantics, making it very suitable for sequence classification tasks such as the one in this study.

Table 5 summarises the training configurations and runtime durations for fine-tuning each transformer model used in this study. All models were fine-tuned on the same dataset using Google Colab with a T4 GPU environment, ensuring consistency in computational resources.

The DistilBERT model, as a lighter version of BERT, had the shortest total fine-tuning duration at approximately 31.5 min for two epochs, confirming its suitability for resource-efficient environments. In contrast, the BERT Base Cased model required the longest time (70.5 min), reflecting its larger parameter set and more complex architecture. The BERT Base Uncased and RoBERTa models had similar fine-tuning durations, around 65 min for two epochs. Although the BERT Base Cased model is generally lighter than RoBERTa, it took longer to train in our experiments. This discrepancy may be due to the specific model implementation or limitations related to the use of free GPU resources, which can affect both training time and performance.

These duration metrics offer practical insights for model selection when balancing performance and fine-tuning cost, especially in environments with limited GPU availability or when rapid iteration is needed.

Table 6 presents the results for accuracy, precision, recall, F1-score, and loss metrics for various transformer-based models used for sentiment prediction, evaluated on both training and test datasets containing only review text.

DistilBERT base uncased demonstrates the best training performance, achieving a training accuracy of 0.93 and the lowest training loss of 0.19, indicating a high degree of learning on the training data. However, on the test set, DistilBERT and BERT base cased both achieve similar test accuracy of over 0.90, with DistilBERT base uncased exhibiting the lowest test loss (0.25), suggesting better generalisation.

The BERT base uncased model follows closely with a test accuracy of 0.89 and a slightly higher test loss of 0.30, indicating slightly less robust generalisation compared to the other version. As capital letters do not have significant meaning in our context, this result is reasonable in practice. Interestingly, RoBERTa, despite achieving a comparable test accuracy of 0.88, shows much weaker performance during training, with a higher training loss (0.25) and lower training accuracy (0.91). This discrepancy could be attributed to the need for more training epochs for the RoBERTa model.

Zooming into the comparison between deep learning and transformer models, the transformer-based models in Table 6 show overall performance that is comparable to or better than the deep learning models presented earlier in Table 7. While the pre-transformer LSTM-based deep learning models achieved a consistent test accuracy of 0.89, the transformer models, such as DistilBERT base uncased and BERT base cased, match or slightly exceed this with a test accuracy of 0.90. Additionally, transformer models demonstrate better optimisation during training than the LSTM-based deep learning models, which had losses ranging from 0.28 to 0.33. This suggests that transformers not only achieve similar levels of accuracy but also generalise better, as indicated by their ability to minimise prediction error.

The hybrid LSTM-CNN deep learning model, while slightly reducing loss in the pre-transformer category, does not achieve the nuanced performance observed in transformer models. This suggests that the robust embeddings inherent in transformer architectures, such as those provided by BERT and RoBERTa, outperform the GloVe and FastText embeddings used in earlier deep learning models. The evaluation highlights the superiority of transformer-based models for sentiment prediction tasks, particularly in their ability to generalise and handle complex textual data more effectively than traditional deep learning approaches.

Overall, these results demonstrate the effectiveness of transformer-based models in sentiment prediction tasks, providing the best balance between training and test performance. The evaluation highlights the importance of proper fine-tuning and hyperparameter optimisation for achieving consistent generalisation across transformer models. Future iterations could explore adjustments to training configurations, particularly for RoBERTa, to improve alignment between training and test metrics.

### 5.4. Entropy-Loss Focused Analysis and Generalisability of the Results

Beyond accuracy and F1-score, understanding how models handle uncertainty is crucial, particularly when dealing with noisy, ambiguous, or sentimentally inconsistent text. Misclassifications offer valuable insights into the entropy-related behaviour of models, highlighting which architectures effectively reduce uncertainty and which struggle to make confident predictions in the face of ambiguity.

We have further analysed the correctly and incorrectly classified comments embedded using FastText and GloVe in the improved stacked LSTM model, which achieved the lowest cross-entropy loss metric with GloVe among the developed pre-transformer deep learning models, indicating more confident and calibrated predictions. Therefore, it is chosen for demonstration purposes. An interesting observation deriven from Figure 12 is that, although the model’s accuracies with both embeddings are the same, the precision value is higher than the recall value when using GloVe (Precision: 0.90; Recall: 0.87), whereas the opposite is true for FastText (Precision: 0.86; Recall: 0.91). This suggests that the model with GloVe embeddings is better at correctly predicting positive cases while minimising false positives, while the model with FastText embeddings is more effective at identifying all actual positive cases, thereby minimising false negatives.

Our experimental data show that FastText produces approximately 35% fewer misclassifications than GloVe for label 0 (negative sentiment). This substantial improvement is due to FastText’s subword-level semantic modelling, which decomposes tokens into character n-grams. This architectural feature enables FastText to preserve semantic meaning even in the presence of lexical noise, such as slang, typographical errors, and inflected or elongated word forms. For example, FastText effectively handles malformed tokens such as “baddd”, “disapointd”, or “nothinggg”, whereas GloVe, lacking subword representation, fails to generalise. Consequently, FastText demonstrates greater fault tolerance and higher recall in noisy textual environments, where lexical variability would otherwise impair traditional word-level models.

However, the results also show that FastText underperforms GloVe for label 1 (positive), misclassifying approximately 80% more samples in this category. We attribute this degradation to FastText’s over-segmentation in cases where tokens are already sentiment-rich and require no further decomposition. Common affective expressions such as “great”, “good”, or “love” are often short and semantically dense. Breaking these tokens into subword units introduces unnecessary granularity, which can dilute their overall sentiment signal. In contrast, GloVe’s holistic word-level embeddings capture these expressions more effectively without semantic fragmentation, leading to better precision and fewer errors in clearly positive texts.

These findings indicate that the effectiveness of subword-level modelling is context-dependent. FastText excels in environments where morphological noise and informal language are prevalent, offering robustness against token irregularities. Conversely, GloVe performs better in syntactically simple, lexically stable cases, where sentiment is expressed through clear and unambiguous tokens that benefit from intact word-level embeddings. By introducing this nuanced analysis, supported by empirical results, we aim to clarify the conditional strengths and weaknesses of each embedding method, thereby strengthening the causal explanation of their observed performance differences.

We then analysed the outputs of the DistilBERT transformer model to compare its performance with the improved stacked LSTM models embedded with GloVe and FastText. The accuracy of the transformer models is higher than that of the pre-transformer deep learning models. Compared to the pre-transformer deep learning models embedded with FastText and GloVe, the DistilBERT model demonstrates superior precision and recall values (Precision: 0.92; Recall: 0.90). Figure 13 below shows the confusion matrix for the performance of the DistilBERT transformer model.

DistilBERT uses contextual embeddings, showing a stronger ability to minimise entropy, especially in complex sentiment cases where textual cues are subtle or mixed (such as sarcastic praise, misaligned ratings, or contradictory language). Its lower cross-entropy loss values (0.25) also support this interpretation, indicating higher model confidence and lower predictive entropy.

This result indicates that the transformer model may predict positive classes with similar accuracy but achieves a higher proportion of correct predictions among all cases identified as positive.

A common limitation when working with a single dataset is the potential for biased or overly specific results. In our study, although we compared a wide range of models and embedding techniques, the original experiments were conducted solely on magazine subscription reviews. This naturally raised questions about the robustness and generalisability of our findings across different domains. To address this concern, we incorporated an additional dataset, the Amazon Gift Card Reviews, from the same 2023 Amazon release. We retrained the best-performing deep learning and transformer models using this new dataset (20,000 reviews balanced between positive and negative sentiments) and compared their performance to the original results.

We observed consistent trends in performance metrics, particularly regarding model ranking and evaluation patterns. This further supports the generalisability and practical applicability of our findings. Table 8 shows the performance of the selected models on the Gift Card dataset. Notably, the DistilBERT base uncased model maintained a similar balance between precision and recall, reflecting the trends observed with the magazine dataset. Compared to Table 6 and Table 7, the results indicate that while the performance pattern is retained, the cross-entropy loss values are consistently lower for all models on the Gift Card dataset, indicating that the models were more confident and less uncertain in their predictions for this domain. This implies that the sentiment signal in Gift Card reviews may be more lexically consistent or less ambiguous, and reinforces the claim that our model evaluations are not narrowly dataset-specific, but rather reflect broader sentiment classification capabilities.

As shown in Figure 14 and Figure 15, the confusion matrices indicate that overall performance trends are consistent with the results obtained for the Magazine dataset. Specifically, DistilBERT maintained high accuracy and a low misclassification rate, particularly for negative reviews. FastText also demonstrated balanced performance, with fewer false positives than GloVe. These outcomes support the broader applicability of our findings across domains and reaffirm the comparative advantages of transformer-based models and subword-level embeddings in handling diverse sentiment expressions.

## 6. Qualitative Analysis of Misclassified Samples and Model Uncertainty

In this section, we provide an in-depth analysis of the misclassified reviews to explore the nature of uncertainty and how different models address it beyond traditional metrics such as accuracy and F1-score. We present a quantified qualitative analysis of the best-performing models: the improved stacked LSTM model with FastText embeddings, the same model with GloVe embeddings, and the transformer-based DistilBERT model, as described in Section 5.4, all trained on the Amazon Magazine dataset. The authors manually annotated the data to identify patterns in the misclassified reviews.

The analysis reveals that different models demonstrate distinct misclassification patterns, reflecting their underlying representational biases and entropy-related behaviour. Table 9 summarises the key causes and relative proportions of errors for each model.

Across all models, the largest group of misclassifications falls into the Other/Subtle Error category, accounting for approximately 30% of the misclassified samples. These include reviews with pragmatic nuances, sarcasm, or ambiguous tones that cannot be easily captured by lexical polarity or syntactic structure. Such cases highlight the intrinsic entropy of natural language, where surface-level polarity features fail to represent deeper semantic intent. DistilBERT showed fewer occurrences in this category compared to LSTM-based models, indicating that contextual embeddings help reduce uncertainty through dynamic semantic adjustment. However, even transformer-based representations struggle with subtle emotional or ironic content, suggesting an upper bound of predictability in sentiment classification tasks.

The second major class of errors concerns Negation Handling, accounting for approximately one-fifth of misclassified instances in FastText- and GloVe-based LSTM models. Phrases containing negators such as “not,” “never,” or “isn’t” often reversed the intended polarity, but static embeddings could not encode this syntactic dependency. This limitation arises from their fixed vector space representations, which lack contextual sensitivity, an issue consistent with high local predictive entropy, where the model’s confidence distribution is nearly uniform across opposing sentiment classes. In contrast, DistilBERT significantly reduced these negation-related misclassifications, demonstrating its ability to propagate context through attention mechanisms that condition each token’s representation on its sentence-level context.

Short and ambiguous comments, such as “Great!” or “Okay,” accounted for about 15% of misclassified reviews for both LSTM models and DistilBERT. These comments typically contain minimal lexical or syntactic information, resulting in sparse feature vectors and high prediction uncertainty. Transformers also showed moderate sensitivity to this issue but tended to predict the dominant sentiment class (positive) with high confidence, indicating overconfidence bias rather than balanced uncertainty calibration. This is consistent with the cross-entropy loss analysis, where lower average loss values in transformers do not necessarily indicate more accurate uncertainty estimation but rather more confident predictions.

Another notable source of misclassification, Sentiment Polarity Misread, was observed across all models, accounting for approximately 14–18% of FastText and GloVe errors and around 16% in the DistilBERT model. These cases typically involve sentences containing highly polarised tokens such as “love”, “great”, or “terrible” used in unexpected or contrastive contexts. For instance, the models frequently misinterpreted comments like “I love how it never works” as positive due to over-reliance on lexical polarity. Such examples reflect high-entropy linguistic constructs, where surface-level sentiment diverges from contextual or pragmatic meaning. While DistilBERT’s contextual embeddings reduced some of these polarity-based errors by incorporating surrounding cues, even transformer-based models continued to struggle with sarcasm, irony, and implicit sentiment—phenomena that require pragmatic and world-level understanding beyond textual semantics.

Contrastive or Mixed Tone sentences (e.g., “It’s good, but not what I expected”) and Token Bias/Weak Context errors further reveal systematic weaknesses in static embeddings. FastText exhibited approximately 11% and 5% errors of these types, respectively. The first reflects an inability to model multi-clause compositional structures, while the latter highlights overreliance on token-level polarity cues rather than discourse-level semantics. DistilBERT outperformed both LSTM variants in these areas, showing improved discourse awareness and contextual weighting, which contribute to its lower entropy in cross-entropy-based uncertainty analysis.

We would also like to draw attention to the size of DistilBERT. Although it is the lightest model among the transformer models fine-tuned in our research, it performed best.

Overall, the comparative examination of misclassifications across models demonstrates that transformer-based models, such as DistilBERT, effectively reduce uncertainty by encoding richer contextual dependencies, yet they still face challenges with high-entropy linguistic phenomena such as irony, sarcasm, or pragmatic ambiguity. These observations complement the quantitative metrics by showing that, while accuracy and loss values provide general performance indicators, entropy-oriented qualitative analysis reveals the latent factors that influence model reliability and interpretability. Future work could extend this analysis with annotation-based calibration studies or ensemble entropy estimation to further disentangle epistemic and aleatoric uncertainty in sentiment prediction tasks.

## 7. Discussion

This study evaluated the performance of traditional machine learning models, pre-transformer deep learning models, and transformer-based models for sentiment analysis, focusing on how different embeddings, architectures, and models affect predictive accuracy. Table 10 compares the best-performing models in each category. The results reveal several important insights into the evolving landscape of natural language processing.

While transformer models achieved the highest accuracy, their cross-entropy loss values were also consistently lower, indicating more confident and calibrated predictions. In contrast, simpler models (e.g., pre-transformer LSTM with static embeddings) showed higher cross-entropy losses, reflecting less certainty, especially on ambiguous reviews. Thus, cross-entropy complements precision and recall by highlighting uncertainty in predictions. In this section, we further discuss these models, focusing on their precision and recall performance.

Transformer-based models, such as DistilBERT and BERT, consistently outperformed traditional and pre-transformer deep learning models, achieving test accuracy above 90%. This superior performance underscores the effectiveness of transformers’ dynamic, context-aware embeddings, which adapt based on word context and enable a nuanced understanding of language. Unlike static embeddings such as GloVe or FastText, transformers excel at handling polysemy and complex semantic relationships, making them ideal for sentiment prediction tasks.

In terms of precision and recall, transformer models again demonstrated superior performance. Both DistilBERT and BERT base cased achieved the highest precision (0.92) among all evaluated models, indicating a strong ability to minimise false positives—particularly valuable in scenarios where misclassification could have reputational or operational consequences, such as product review filtering for sensitive brands. Moreover, their recall scores (0.90 for DistilBERT and 0.89 for BERT) reflect their robustness in correctly identifying a high proportion of sentiment-bearing instances, whether positive or negative.

This level of sensitivity is especially significant when compared to models such as Random Forest with Bag of Words and the Hybrid LSTM-CNN with FastText, both of which showed lower recall values (0.85). These results highlight their tendency to miss relevant sentiment cues, potentially overlooking crucial feedback. In contrast, transformer models achieve a well-balanced trade-off between precision and recall, enhancing their suitability for both high-stakes decision-making and scalable sentiment analysis tasks.

However, this performance comes at a computational cost. The high computational demands required reducing the dataset from 25,000 to 20,000 samples to enable training within the free GPU quota provided by Google Colab. While this reduction made practical implementation possible, it may have limited the generalisability of the findings, particularly for transformer models, which typically benefit from larger datasets. Additionally, it affected accuracy performance. We observed that the model built with the 25,000-sample dataset (20,000 for training, 5000 for testing) achieved 1% higher accuracy at the end of the first epoch than the model built with the 20,000-sample dataset (16,000 for training, 4000 for testing). This suggests that our transformer models could perform even better with larger datasets.

Traditional machine learning models such as Random Forest (with hyperparameter tuning improving its performance by approximately 1%) and Logistic Regression showed competitive results when combined with BoW embeddings and additional features, achieving accuracies of up to 89%. Additionally, FastText also consistently outperformed GloVe in this context, likely because it captures subword-level information, which aids in handling rare or misspelt words. Despite their competitive performance, traditional models depend heavily on supplementary features, such as helpful votes, to enhance accuracy, a reliance that modern transformers inherently reduce through their embedding mechanisms.

Pre-transformer deep learning models, particularly the improved stacked LSTM and the hybrid LSTM-CNN, served as a middle ground in both performance and computational efficiency. Both models achieved an accuracy of 0.89, with similar F1-scores. Notably, the hybrid LSTM-CNN model using FastText showed a high precision of 0.90 but a lower recall of 0.85. This imbalance suggests that the model is conservative in assigning sentiment labels but may miss relevant cases, which could be problematic for tasks requiring high sensitivity. In contrast, the improved stacked LSTM model with GloVe embeddings maintained a more balanced performance (precision 0.90, recall 0.87), indicating a better trade-off between minimising false positives and capturing true sentiments.

These precision-recall patterns illustrate how different models prioritise either conservative prediction (high precision, low recall) or comprehensive detection (high recall, potentially lower precision). In real-world deployments, this trade-off must align with application requirements. For instance, high recall is essential for detecting negative reviews in customer service triage, whereas high precision may be prioritised in automated moderation to avoid false positives.

Beyond performance, the computational trade-offs present significant challenges. Although transformers achieved the highest accuracy, their fine-tuning required substantial computational resources. Lightweight transformer models such as DistilBERT offer a promising approach to maintaining high performance while reducing computational costs. Further research is needed to address dataset limitations caused by resource constraints, ensuring richer training data without compromising hardware compatibility.

In conclusion, the practical implications of these findings go beyond academic benchmarking. The consistent superiority of transformers demonstrates their potential for real-world sentiment analysis applications, such as customer feedback systems and social media monitoring. Meanwhile, the strong performance of traditional models with advanced embeddings highlights their relevance in resource-limited environments, emphasising the importance of tailoring solutions to specific application needs.

## 8. Conclusions and Future Work

This study presents a comprehensive comparison of traditional machine learning models, pre-transformer deep learning models, and transformer-based models for sentiment prediction, offering valuable insights into their respective strengths and limitations. Transformer models such as DistilBERT and BERT achieved the highest accuracy (over 90%), demonstrating their ability to dynamically adapt embeddings to word context, making them highly effective for nuanced language tasks. Traditional models like Random Forest and SVM performed competitively with advanced embeddings such as FastText, demonstrating their continued relevance in resource-constrained environments. Pre-transformer models, such as LSTM-based deep learning models, bridged the gap but ultimately lacked the generalisation capabilities of transformers.

The study also highlights the critical role of embedding methods. FastText outperformed GloVe in traditional models, emphasising the importance of subword-level information for handling rare or domain-specific words. The inclusion of supplementary features improved traditional model performance, but these are not used in transformer models; only the reviews had an impact, as they inherently capture rich contextual information through their attention mechanisms.

While reducing the dataset from 25,000 to 20,000 samples enabled fitting the transformer models on limited hardware, it may have constrained the models’ generalisability. These findings highlight the importance of balancing computational feasibility with dataset richness for optimal model performance. We sought to address this challenge by using a lightweight DistilBERT model. Other lightweight models, such as ALBERT or TinyBERT, can also be considered.

Considering these aspects, we observe that the ability to reduce uncertainty (i.e., entropy minimisation) is closely linked to the model’s representational power and contextual sensitivity. This analysis shows that while simpler models may offer computational efficiency, they often struggle to resolve uncertainty in real-world language, where sentiment is rarely binary or explicit.

We note a growing trend in the field towards integrating transformer models in multimodal applications. We suggest that future work could explore newer transformer architectures in conjunction with Amazon’s text reviews, as well as accompanying visual and video-based feedback, to fully leverage the capabilities of multimodal sentiment analysis models.

Future work should explore not only overall performance measures but also deeper information-theoretical metrics, such as prediction entropy, KL divergence, and mutual information, and may incorporate entropy regularisation to improve robustness, particularly in ambiguous cases.

In conclusion, this work advances the understanding of sentiment analysis methodologies, demonstrating the transformative potential of modern transformers while highlighting the relevance of traditional models in specific contexts. By addressing computational challenges and exploring new domains, future research can further enhance the applicability and impact of these findings in the field of natural language processing.

## Figures and Tables

**Figure 1 entropy-27-01202-f001:**
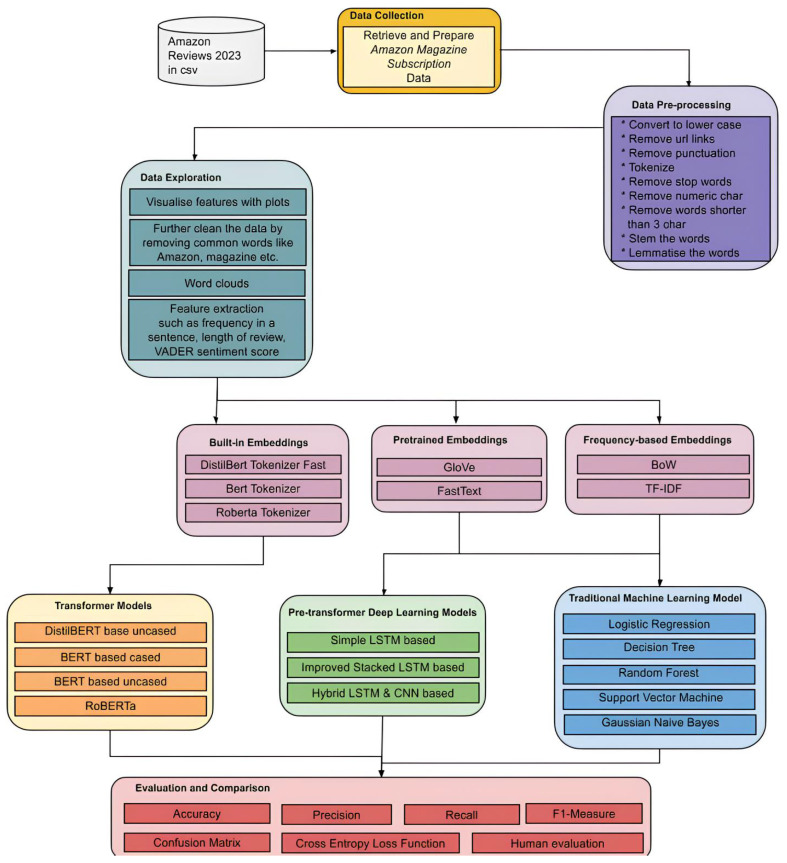
Depict of methodological design.

**Figure 2 entropy-27-01202-f002:**
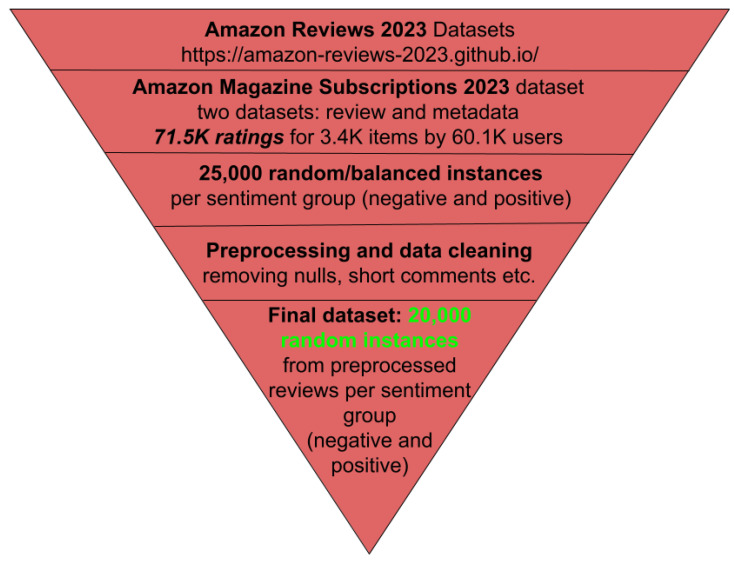
Depict of selecting and preparing data samples for sentiment analysis from the dataset.

**Figure 3 entropy-27-01202-f003:**
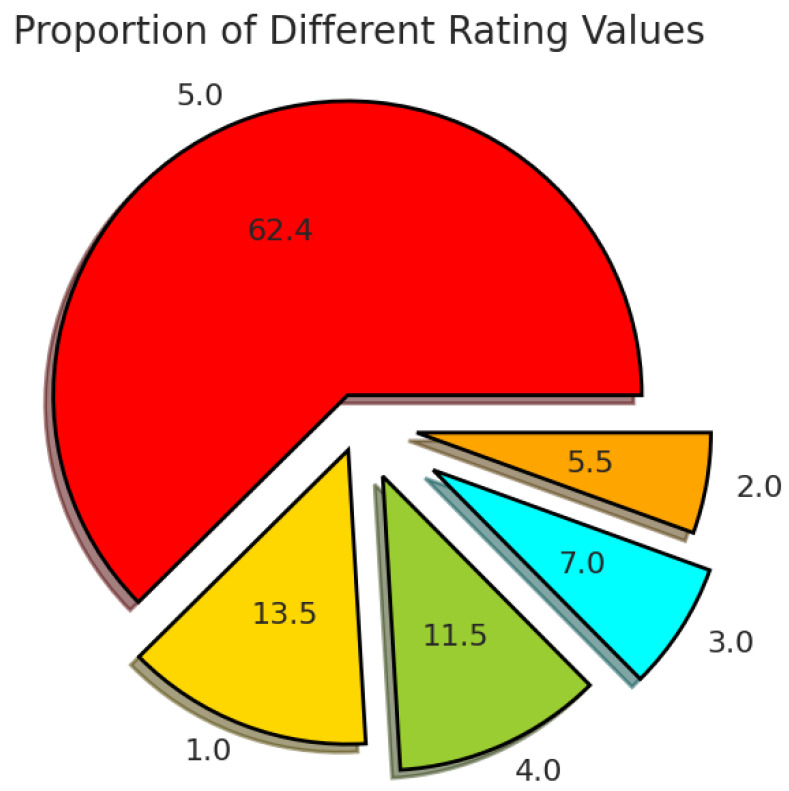
Proportion of different ratings.

**Figure 4 entropy-27-01202-f004:**
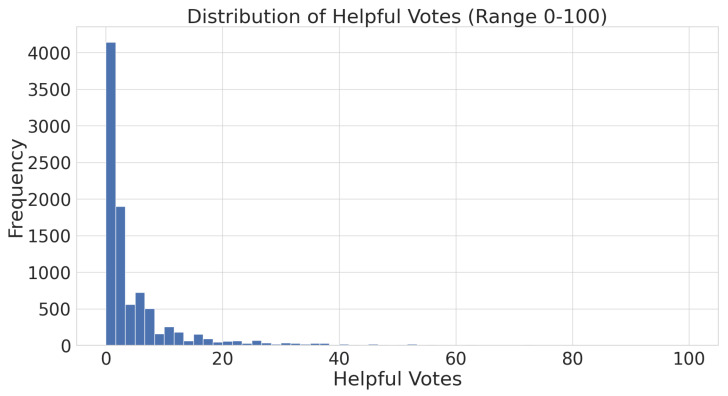
Distribution of the helpful votes across reviews.

**Figure 5 entropy-27-01202-f005:**
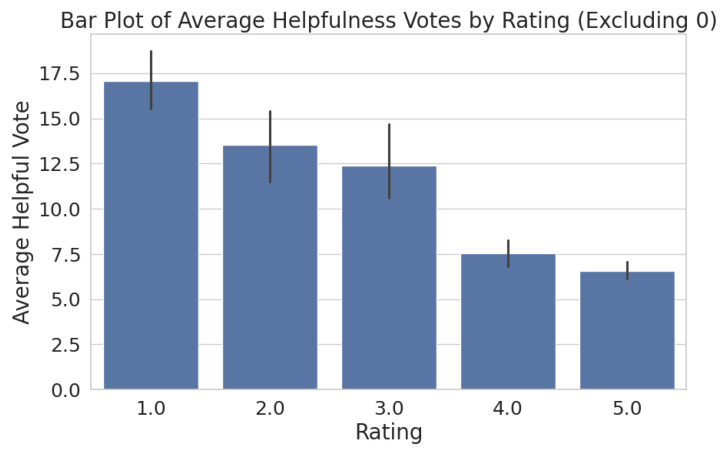
Average helpfulness votes received by reviews for each rating (1 to 5 stars), excluding reviews with zero helpful votes.

**Figure 6 entropy-27-01202-f006:**
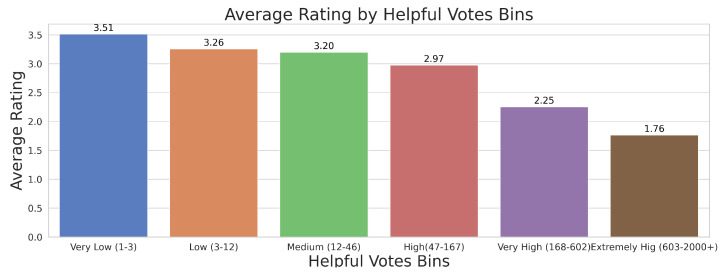
Average rating by helpful votes bins.

**Figure 7 entropy-27-01202-f007:**
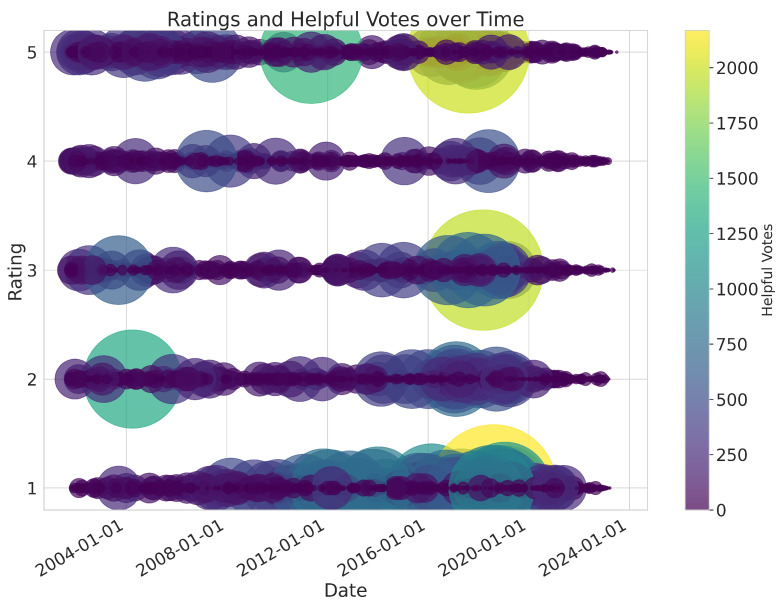
Bubble chart for general trends to identify outliers in ratings and helpful votes over time.

**Figure 8 entropy-27-01202-f008:**
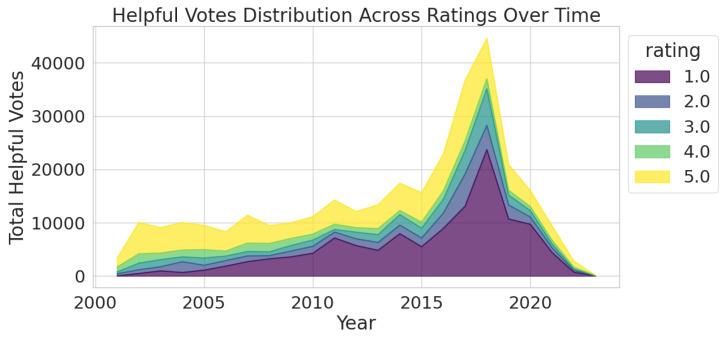
Stacked Area Chart for cumulative trends in the distribution of ratings and helpful votes over time.

**Figure 9 entropy-27-01202-f009:**
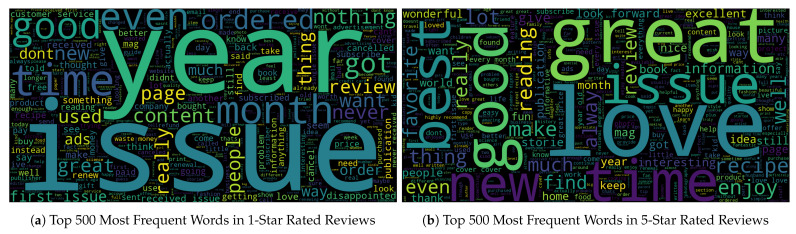
Word Clouds of Most Frequently Used Words in (**a**) 1-Star and (**b**) 5-Star Product Reviews.

**Figure 10 entropy-27-01202-f010:**
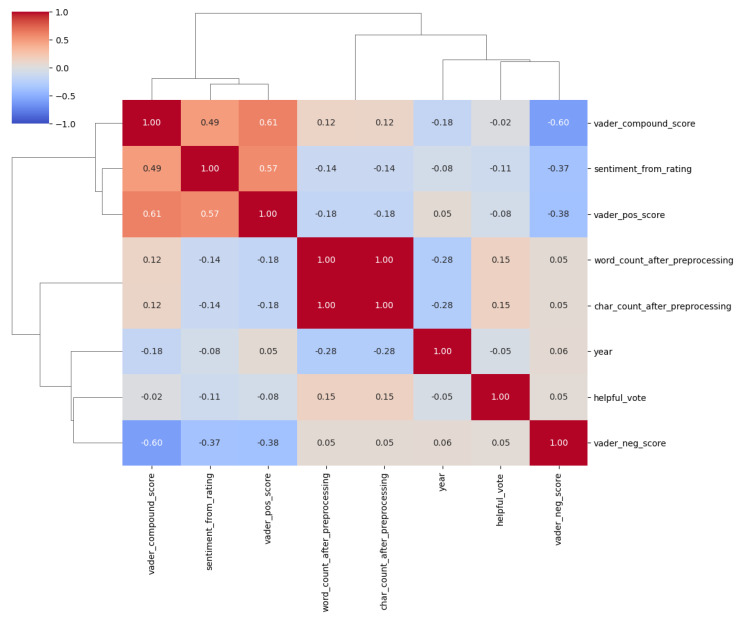
Clustered heatmap representation of the correlated features.

**Figure 11 entropy-27-01202-f011:**
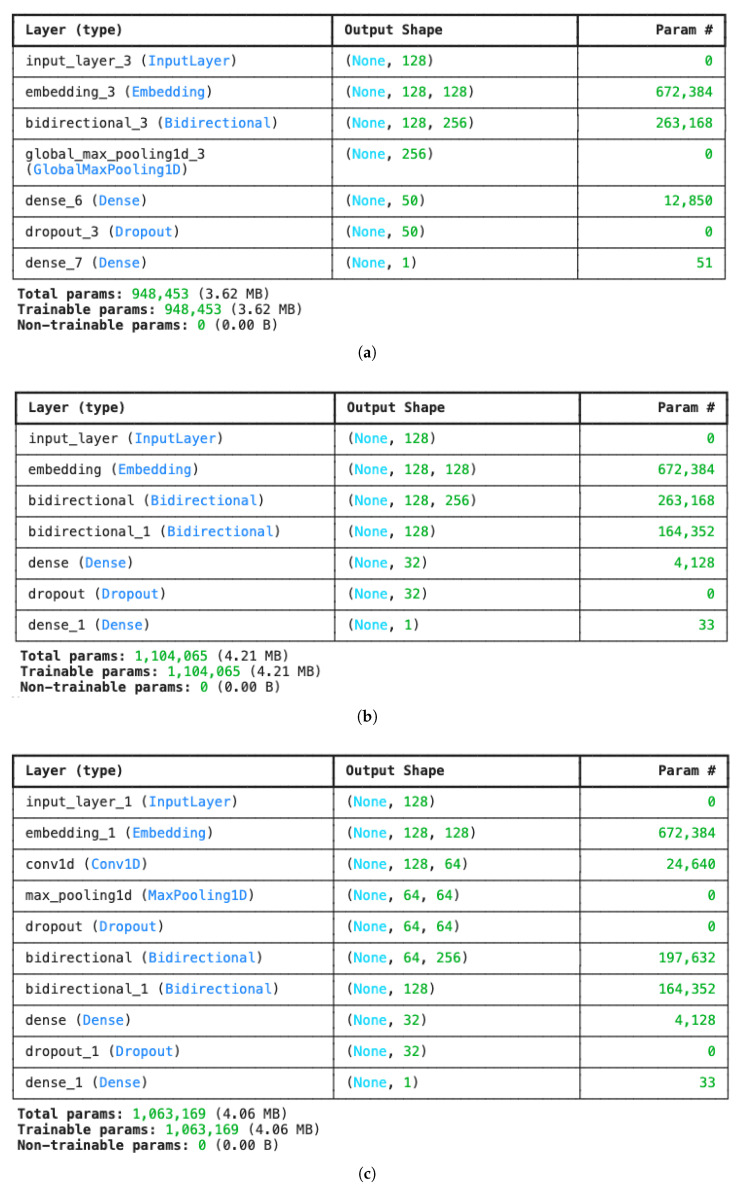
Architectures of pre-transformer sequential deep learning models developed in this research, including layer configuration, size, and parameter settings. (**a**) Developed LSTM-based simplest deep learning (Model 1—Simple LSTM based); (**b**) Developed improved LSTM-based deep learning (Model 2—Improved Stacked LSTM based); (**c**) Developed hybrid deep learning (Model 3—Hybrid LSTM-CNN based).

**Figure 12 entropy-27-01202-f012:**
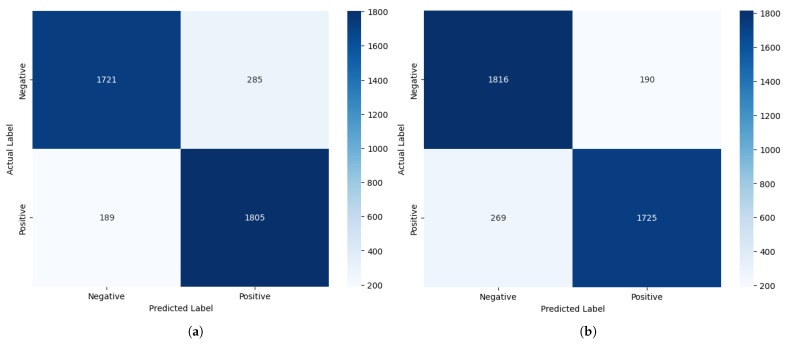
Comparison of Confusion Matrices for Improved Stacked LSTM (Model 2) with Different Embedding Models. (**a**) Confusion Matrix: Improved Stacked LSTM Model embedded with FastText; (**b**) Confusion Matrix: Improved Stacked LSTM Model embedded with GloVe.

**Figure 13 entropy-27-01202-f013:**
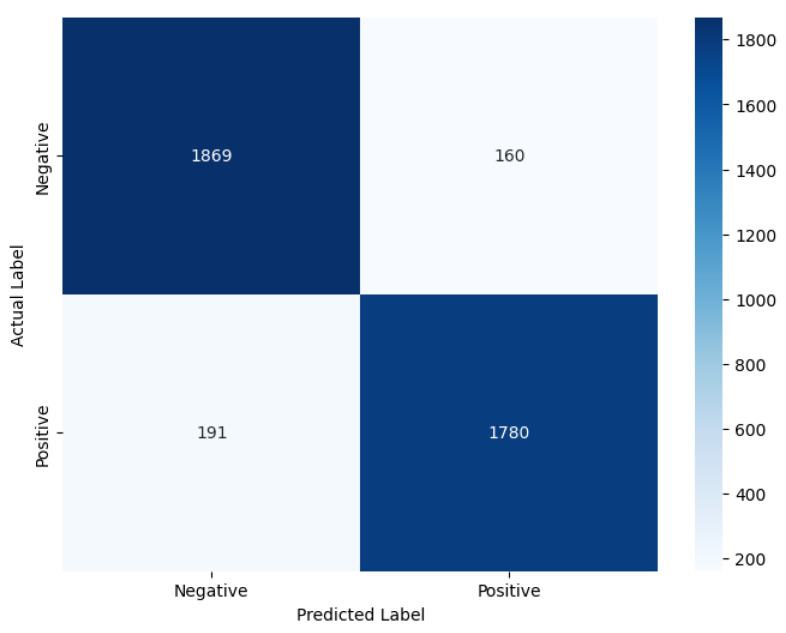
Confusion Matrix: DistillBERT transformer model.

**Figure 14 entropy-27-01202-f014:**
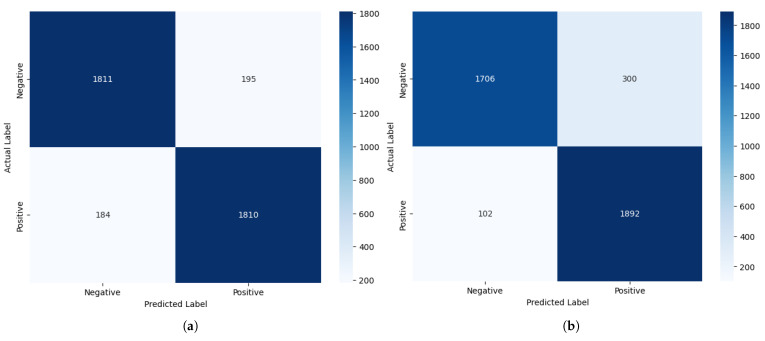
Comparison of Confusion Matrices for Improved Stacked LSTM (Model 2) with Different Embedding Models (Gift Cards). (**a**) Confusion Matrix: Improved Stacked LSTM Model embedded with FastText (Gift Cards); (**b**) Confusion Matrix: Improved Stacked LSTM Model embedded with GloVe (Gift Cards).

**Figure 15 entropy-27-01202-f015:**
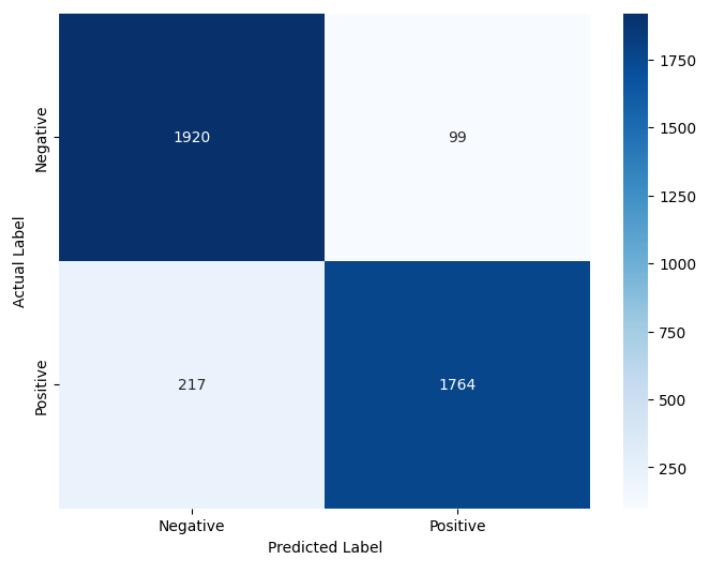
Confusion Matrix: DistillBERT transformer model (Gift Cards).

**Table 1 entropy-27-01202-t001:** Comparison of Embedding Methods Experimenting with Traditional Machine Learning Models: Vector Size and External Feature Count.

Embedding Method	Text Vector Size	External Features
FastText	128	4
GloVe	100	4
TF-IDF	10,386	4
Bag of Words	10,386	4

**Table 2 entropy-27-01202-t002:** Accuracy and F1-Score of Traditional Machine Learning Models with No Embedding and Static Embedding Methods.

Model	No Embeddings		Bag of Words (BoW)				TF-IDF			
	Acc.	F1	Acc. Reviews Only	F1 Reviews Only	Acc. +Feat.	F1 +Feat.	Acc. Reviews Only	F1 Reviews Only	Acc. +Feat.	F1 +Feat.
Logistic Regression	0.62	0.69	0.89	0.89	0.89	0.89	0.89	0.89	0.89	0.89
Decision Tree	0.63	0.65	0.80	0.80	0.79	0.79	0.80	0.80	0.79	0.79
Random Forest	0.69	0.69	0.89	0.88	0.89	0.88	0.88	0.87	0.88	0.87
SVM	0.61	0.69	0.88	0.88	0.88	0.88	0.87	0.87	0.88	0.88
Gaussian Naive Bayes	0.56	0.68	0.63	0.70	0.63	0.70	0.66	0.71	0.66	0.71

**Table 3 entropy-27-01202-t003:** Accuracy and F1-Score of Traditional Machine Learning Models with Pre-Trained Embedding Methods.

Model	GloVe				FastText			
	Acc. Reviews Only	F1 Reviews Only	Acc. +Features	F1 +Features	Acc. Reviews Only	F1 Reviews Only	Acc. +Features	F1 +Features
Logistic Regression	0.84	0.84	0.84	0.84	0.88	0.87	0.88	0.88
Decision Tree	0.74	0.74	0.74	0.74	0.78	0.77	0.77	0.77
Random Forest	0.84	0.83	0.84	0.84	0.87	0.86	0.87	0.87
SVM	0.85	0.84	0.84	0.84	0.88	0.87	0.88	0.88
Gaussian Naive Bayes	0.70	0.64	0.73	0.68	0.75	0.71	0.77	0.73

**Table 4 entropy-27-01202-t004:** Configuration and Training Time of Pre-Transformer DL Models with GloVe and FastText Embeddings (5 Epochs, T4 GPU).

Model	Embedding	Params (MB)	Total Time	Per-Epoch Time
DL Model-1 (LSTM)	GloVe (Trainable)	2.25 M (8.5)	22 min	4 min + 20 s
DL Model-2 (Stacked LSTM)	GloVe (Trainable)	2.43 M (9.17)	26.5 min	5 min + 20 s
DL Model-3 (LSTM-CNN)	GloVe (Trainable)	2.39 M (9.10)	14 min	2.5–3 min
DL Model-1 (LSTM)	FastText (Frozen)	0.95 M (3.62)	25.7 min	5–5.25 min
DL Model-2 (Stacked LSTM)	FastText (Trainable)	1.10 M (4.21)	35.2 min	7–7.15 min
DL Model-3 (LSTM-CNN)	FastText (Trainable)	1.06 M (4.06)	17.5 min	3.5 min

**Table 5 entropy-27-01202-t005:** Transformer Models—Tokenization Method and Fine-Tuning Time in Minutes (T4 GPU, 2 Epochs).

Model	Tokenizer	Total Time	Time Per Epoch
DistilBERT Base Uncased	DistilBERT Tokenizer (Fast)	31.51	15.75
BERT Base Cased	BERT Tokenizer (‘bert-base-cased’)	70.52	35.26
BERT Base Uncased	BERT Tokenizer (‘bert-base-uncased’)	64.72	32.36
RoBERTa	RoBERTa Tokenizer (Fast)	65.80	32.90

**Table 6 entropy-27-01202-t006:** Performance Metrics for Train and Test on Transformer Models.

Transformer Models	Train (Reviews Only)		Test (Reviews Only)				
	Acc.	Loss	Acc.	Precision	Recall	F1	Loss
DistilBERT base uncased	0.93	0.19	0.92	0.92	0.90	0.91	0.25
BERT base cased	0.92	0.21	0.90	0.92	0.89	0.90	0.29
BERT base uncased	0.93	0.20	0.89	0.85	0.95	0.89	0.30
RoBERTa	0.91	0.25	0.88	0.83	0.94	0.88	0.33

**Table 7 entropy-27-01202-t007:** Performance Metrics of Pre-Transformer Sequential Deep Learning Models for Sentiment Prediction with GloVe and FastText Embeddings (5 epochs).

Pre-Transformer Models	GloVe (Reviews Only)					FastText (Reviews Only)				
	Accuracy	Precision	Recall	F1-Score	Loss	Accuracy	Precision	Recall	F1-Score	Loss
Simple LSTM based model 1	0.89	0.89	0.89	0.89	0.33	0.89	0.88	0.91	0.89	**0.31**
Improved Stacked LSTM model 2	0.89	0.90	0.87	0.88	**0.31**	0.89	0.86	0.91	**0.89**	0.33
Hybrid LSTM-CNN model 3	0.89	0.87	0.91	0.88	0.32	0.89	0.90	0.85	**0.89**	**0.28**

**Table 8 entropy-27-01202-t008:** Use of the Amazon Gift Card Dataset in Best-Performing Sentiment Prediction Models.

Models	Embedding	Accuracy	Loss	F1	Precision	Recall
Improved Stacked LSTM #2	GloVe	0.90	0.14	0.90	0.86	0.95
Improved Stacked LSTM #2	FastText	0.90	0.16	0.90	0.90	0.91
DistilBERT base uncased	DistillBERT Tokenizer Fast	0.92	0.16	0.92	0.95	0.89

**Table 9 entropy-27-01202-t009:** Error-type analysis of misclassified reviews across models using the Magazine dataset.

Cause/Type	Quantitative Indicator (Heuristic)	Distribution (Model-Wise)	Model	Interpretation
Other/Subtle Error	Does not match any strong heuristic, likely pragmatic nuance, sarcasm, or unclear tone	∼32% (FastText), ∼28% (GloVe), ∼9% (DistilBERT)	All models	This is the largest category, reflecting the difficulty of handling subtle sentiment cues.
Negation Handling	Contains negations like “not”, “never”, “no”, “isn’t”, “wasn’t”	∼21% (FastText), ∼22% (GloVe)	FastText, GloVe	Static embeddings struggle with handling the polarity shift induced by negation.
Short/Ambiguous	1–3 word reviews or vague language (“good”, “okay”)	∼15% (FastText), ∼15% (GloVe), ∼15% (DistilBERT)	All models	Short reviews lack context; the model guesses sentiment from weak signals.
Sentiment Polarity Misread	Contains strong polarity words like “great”, “horrible”, “love”, etc.	∼14% (FastText), ∼18% (GloVe), ∼16% (DistilBERT)	All models	Words with fixed sentiment can mislead models in sarcastic or ironic contexts.
Contrastive/Mixed Tone	Phrases like “but”, “however”, “although” exist	∼11% (FastText)	FastText	Model fails to track sentence structure beyond a single clause.
Token Bias/Weak Context	Low lexical variety and low content length	∼5% (FastText)	FastText	Overreliance on individual tokens like “great” or “happy”.

**Table 10 entropy-27-01202-t010:** Comparison of Best Performing Models Across Categories for Sentiment Prediction with Amazon Magazine Dataset.

Models	Embedding	Accuracy	Loss	F1	Precision	Recall
**Traditional Machine Learning**						
Logistic Regression	Bag of Words	0.89	-	0.89	0.88	0.90
Random Forest	Bag of Words	0.89	-	0.88	0.92	0.85
**Pre-Transformer**						
Improved Stacked LSTM #2	GloVe	0.89	0.31	0.88	0.90	0.87
Hybrid LSTM-CNN #3	FastText	0.89	0.28	0.89	0.90	0.85
**Transformer**						
DistilBERT base uncased	DistillBERT Tokenizer Fast	0.92	0.25	0.91	0.92	0.90
BERT base cased	BERT Tokenizer	0.90	0.29	0.90	0.92	0.89

## Data Availability

The data is publicaly available on this link: https://amazon-reviews-2023.github.io/ (accessed on 19 November 2025).

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
