# Peer review of "Transformer and Pre-Transformer Model-Based Sentiment Prediction with Various Embeddings: A Case Study on Amazon Reviews"

_entropy, 2025, doi:10.3390/e27121202_

Round 1
Reviewer 1 Report
Comments and Suggestions for Authors
- The literature analysis and summary of sentiment analysis are inadequate. The section on sentiment analysis needs to be strengthened, and it is recommended to add analyses of influential papers, such as the one with the DOI: https://doi.org/10.1016/j.ipm.2023.103354.
- The paper only compares and evaluates existing methods, and the innovation of the paper itself is not reflected.
- The processing logic for "sentiment-rating mismatches" is not explained: although this scenario is mentioned as a challenge, details such as "how to define such samples in the dataset" (e.g., through manual annotation or automatic algorithm identification) and "differences in how various models handle these samples" (e.g., whether Transformers are more adept at addressing mismatches than traditional machine learning) are not supplemented. This weakens the paper’s support for "adaptation to complex scenarios"; it is recommended that the authors analyze the literature with the DOI: https://doi.org/10.1016/j.ipm.2021.102818.
- The causal reasoning is somewhat weak: while it is mentioned that "FastText outperforms GloVe due to its subword-level semantic richness," specific supporting data is lacking (e.g., in the recognition of a certain type of sentiment label, the recall rate of FastText is X% higher than that of GloVe, and by how much the subword representation improves the fault tolerance for slang/spelling errors). The persuasiveness of the causal relationship needs to be enhanced.
- The representativeness of the dataset is questionable: the limitations of the "Amazon Magazine Subscriptions 2023 dataset" are not analyzed—such as the singularity of text types (only magazine subscription reviews) and domain bias (e.g., bias toward specific consumer groups or review styles). It remains unclear whether the conclusions will hold if other domain datasets (e.g., social media text, reviews of electronic products/food) are used. An analysis of "the impact of dataset bias on results" needs to be added to avoid confining the conclusions to a single scenario.
- The comparison of embedding methods is incomplete: among static embeddings, only GloVe and FastText are compared, with classic methods excluded; among Transformer embeddings, model scales are not distinguished (e.g., BERT-base vs. BERT-large, DistilBERT vs. MobileBERT). This makes it impossible to reflect the trade-off relationship between "model size, performance, and computational cost," limiting the generalizability of the conclusions.
Reviewer 2 Report
Comments and Suggestions for Authors
Ad-hoc Dataset Reduction: A significant concern is the reduction of the dataset from 25,000 to 20,000 instances due to the computational limits of a free GPU service. This is not a principled decision based on experimental design but rather a hardware limitation. The authors themselves note that a larger dataset improved accuracy by 1% even in the first epoch . This compromise undermines the generalizability of the findings, as transformer models, in particular, are known to benefit from larger datasets.
Simplification of the Problem: The decision to exclude 3-star (neutral) reviews simplifies the sentiment analysis task from a more challenging multi-class problem to a binary one. The justification provided is that including a neutral class "significantly declined" the models' performance. However, this is a circular argument; the task was made easier to achieve better results. A stronger manuscript would have analyzed why models struggle with neutral reviews and discussed the implications of ignoring this ambiguous but significant category of user feedback.
Inconsistent Use of Features: The study uses extracted features (like helpful_vote and VADER scores) to boost the performance of traditional machine learning models. However, these features were not incorporated when training the transformer models, which only used the review text. This creates an inconsistent comparison framework. The performance gap between the paradigms might be different if all models were given access to the same feature sets.
The paper claims to contribute to "sustainable AI" by using a smaller dataset and a lightweight model like DistilBERT. However, this was driven by resource constraints rather than a deliberate research goal to investigate efficiency. The framing feels opportunistic and detracts from the core sentiment analysis comparison.
The "Gap in the Literature": The authors claim that the comparative effectiveness of different embeddings in transformer architectures is "underexplored" and that few studies "systematically compare all three [paradigms] within a unified evaluation framework". While this specific combination of models on the 2023 Magazine dataset may be new, the broader concept of comparing traditional ML, deep learning, and transformers for sentiment analysis is a well-established area of research. The novelty is narrower than suggested.
Underutilized Cross-Entropy Metric: The manuscript highlights categorical cross-entropy as an "important evaluation metric" to capture model uncertainty. While the loss values are reported in tables, the analysis remains shallow. The discussion primarily states that lower loss is better and indicates more confident predictions.
Anecdotal Error Analysis: The section on misclassifications (5.4) provides interesting qualitative examples , which is a strength. However, the analysis is anecdotal and lacks quantitative rigor. For instance, it observes that short sentences are often misclassified but does not quantify this relationship. It also "suspects" that a preprocessing step ("I'll" becoming "ill") caused an error, which points to a potential flaw in their own data preparation pipeline rather than an insightful model limitation.
Reviewer 3 Report
Comments and Suggestions for Authors
This paper systematically compares the performance of sentiment classification models across various architectural approaches, including traditional machine learning and Transformer-based models, while evaluating the effectiveness of multiple embedding techniques. The study further introduces classification cross-entropy as an evaluation metric to capture prediction confidence and model uncertainty. Ultimately, by proposing a multi-level framework integrating information-theoretic assessment, it provides valuable insights for the sentiment analysis domain, facilitating more rational and efficiency-focused model selection in practical applications. However, the paper still has room for improvement in the following areas, with specific issues and suggestions outlined below.
- Many studies has already addressed this topic, what is the difference or contribution of this study?
- The literature review section lacking systematic overview the classification methods of sentiment classification, as well as the current application status of information-theoretic metrics like cross-entropy and predictive entropy in sentiment analysis uncertainty assessment. The content of literature review section is also hard to understand.
- This paper does not incorporate newer methods emerging in the field of sentiment analysis in recent years. It is recommended to include newer model approaches for supplementary validation to enhance the timeliness and persuasiveness of the research conclusions.
- The dataset used exclusively Amazon magazine subscription review data. However, certain models may perform better on domain-specific datasets. Results from a single dataset may not fully reflect a model's generalizability, potentially skewing experimental conclusions. It is recommended to supplement with sentiment analysis datasets from other domains for comparative experiments, thereby more thoroughly validating the paper's claims and conclusions.
- Regarding dataset processing, the paper mentions reducing the dataset from 25K to 20K to balance computational efficiency and performance, but does not specify how this reduction was achieved. If it involved simply removing some data, is this approach reasonable? Could it compromise the validity of the results?
- Does the reduced dataset ensure consistent positive and negative sample distributions with the original dataset? If the data distribution of the original dataset is disrupted, could this reduction affect the experimental results?
- The paper introduces classification cross-entropy to capture model uncertainty but does not conduct an in-depth analysis of the specific causes of this uncertainty. It also lacks comparisons of how different models handle texts with higher uncertainty—for example, whether Transformer models are more effective than traditional machine learning models at reducing prediction uncertainty for specific text types. We recommend supplementing the paper with corresponding comparative experiments to further deepen the application value of information-theoretic evaluation and enhance the analytical depth of the research.
Round 2
Reviewer 1 Report
Comments and Suggestions for Authors
The commments are revised basically.
The conclusion section is excessively lengthy; it is recommended to be more concise.
Author Response
Thank you very much for your time and constructive feedback. Your comments have been invaluable in improving the quality of our paper. We are pleased that you found our revisions satisfactory. As part of the final refinements, we have revised the language throughout the manuscript for clarity. We have also shortened the Conclusion by removing less essential commentary, such as: "As authors of this manuscript, we also value the extended work on domain-specific corpora (e.g., medical or legal reviews) and multilingual or under-represented languages sentiment datasets to assess the adaptability of these models."

Reviewer 2 Report
Comments and Suggestions for Authors
The authors have made important improvements in the manuscript.
Author Response
Thank you very much for your time. Your comments have been invaluable in enhancing the quality of our paper. We are glad that you have found our improvements sufficient.
Reviewer 3 Report
Comments and Suggestions for Authors
In this version, the authors have made revisions to address the previous review comments. The overall effort is substantial, particularly with the addition of the Amazon gift card review dataset, the inclusion of misclassification sample studies, and the refinement of the literature review framework. These enhancements effectively address some of the earlier concerns.
However, the research's innovation remains limited. Its core contributions center on the integration and application of evaluation frameworks, without proposing breakthrough methods in model architecture or core methodologies. While multidimensional analysis enhances the study's value, its level of innovation still has room for improvement. Additionally, could the current article structure be further optimized? For instance, although the literature review section now includes the evolution of methods, the analysis of the current application status of information-theoretic metrics could be elevated to a separate subsection. This would strengthen the logical hierarchy and make the overall structure clearer and more accessible.
Author Response
Thank you, as always, for your time and constructive feedback. We are pleased that you appreciated our efforts to improve the overall quality of the manuscript.
We understand your concerns regarding the novelty of the paper in the absence of a newly proposed model. However, we believe that conducting a systematic and comparative methodological evaluation of existing frameworks with different embedding methods remains a valuable contribution to the field. In particular, based on your earlier feedback, we introduced a systematic qualitative analysis of misclassified samples through the lens of model uncertainty. We consider this addition a meaningful novel contribution that complements the quantitative findings and enhances interpretability.
That said, we fully agree that proposing a new model specifically designed to address the identified uncertainty categories would represent an important and impactful follow-up study.
To address your comments regarding the structure and clarity of the literature review, we have added Section 2.5: Information-Theoretic Metrics in Sentiment Analysis, which outlines the role of entropy-based and information-theoretic evaluation metrics in sentiment classification. We hope that this structural revision improves the logical flow of the literature review and strengthens the paper’s conceptual grounding.
All of these improvements are highlighted in the latest version of the manuscript. We sincerely hope they meet your expectations and contribute to the manuscript's overall quality.
